# HIERARCHICAL OVERLAPPING CLUSTERING: COST FUNCTION, ALGORITHM AND SCALABILITY

## ABSTRACT

Overlap and hierarchy are two prevalent phenomena in clustering, and usually coexist in a single system. There are several studies on each of them separately, but it is unclear how to characterize and evaluate the hybrid structures yet. To address this issue, we initiate the study of hierarchical overlapping clustering on graphs by introducing a new cost function for it. We show the rationality of our cost function via several intuitive properties, and develop an approximation algorithm that achieves a provably constant approximation factor for its dual version. Our algorithm is a recursive process of overlapping bipartition based on local search, which makes a speed-up version of it extremely scalable. Our experiments demonstrate that the speed-up algorithm has good performances in both effectiveness and scalability on synthetic and real datasets.

## 1 INTRODUCTION

Clustering is a major task in data mining and has a wide range of applications in many areas. Two fundamental categories of clustering have attracted in-depth study recently. The first is hierarchical clustering (HC) which requires a recursive partitioning of a graph into smaller clusters to form a cluster tree Dasgupta (2016); Li & Pan (2016); Cohen-Addad et al. (2019); Charikar & Chatziafratis (2017); Moseley & Wang (2017); Naumov et al. (2021). The other is overlapping clustering (OC) that allows data points to belong to multiple clusters Orecchia et al. (2022); Zhang et al. (2007); Shen et al. (2009); Chen et al. (2010); Nicosia et al. (2009); Whang et al. (2016); Li et al. (2017); Yang & Leskovec (2012a). These two structures are widely present in the real world, and the hybrid structure of hierarchical overlapping clustering (HOC) that allows for the presence of overlaps among hierarchical clusters better reflects real-world scenarios. For instance, in social networks, an agent may belong to several different groups, which can form larger communities with overlapping structures based on different themes. In cooperation networks, the coauthors of a paper can be thought of as a small cluster, which may belong to more than one research area due to the topic. This hybrid structure in fact poses a significant challenge to the study of clustering. There are many works for HC and OC separately, but we lack research on HOC. In this paper, we address this problem.

We study HOC on graphs. Constructing a cost function is a common method for the research on HC and OC. Similarly, a proper cost function is helpful to evaluate the quality of HOC, which transforms the HOC problem to an optimization task. In this paper, we propose a new cost function for HOC, and present an approximation algorithm for it in some reasonable condition. Our contributions are summarized as follows:

(1) **Cost function.** We propose a cost function (Definition 2.8) that is the first one for HOC to our best knowledge. The cost function is evaluated on overlapping clustering graphs, and can be unified with Dasgupta's cost function for HC trees in the specific case of non-overlap. We give a comprehensive study on the rationality of this cost function from multiple perspectives such as examples, algorithms, experiments, and a series of properties including compatibility (Property 2.11), additivity of nodes (Property 2.12) and binary optimality (Property 2.13).

(2) **Approximation algorithm.** Based on our cost function, we formulate the primal and the dual versions of HOC, respectively. We provide an $a = \frac{2}{3\sqrt{6}} - \Theta(\frac{1+\epsilon}{n})$-approximation algorithm (Algorithm 2) for the dual $k$-HOC problem, where $k \in \mathbb{Z}^+$ is an upper bound of key clusters (explained in Definition 2.10). Our algorithm is a recursive process of overlapping bipartition in

which the height of the overlapping clustering graph and the cluster number are both restricted to two. We denote this simple case by 2-OC, which is the theme of recent study of Orecchia et al. (2022) on OC. We show that our algorithm also achieves an approximation factor $(1-a)(1+d_{max}/d_{avg})$ for the primal 2-OC problem. Instead of the complicated "cut-matching and improve" approach in Orecchia et al. (2022), our method for 2-OC takes a simple local search heuristic based on our cost function, which makes our algorithm much more scalable.

(3) **Effectiveness and scalability.** We speed up our approximation algorithm by some simple heuristics during local search, and verify its effectiveness and scalability by experiments. For effectiveness, experimental results demonstrate that on random graph models with good clustering structures, our algorithm is able to reconstruct the overlapping clusters. For scalability, benefiting from our subtle design of cost function and simple local search process, on real datasets with around one million vertices and three million edges, the runtime of our speed-up algorithm implemented on a single laptop is less than 12 minutes, which is only around 20% of the runtime of the baseline method that runs on a server.

## 1.1 RELATED WORK

**Hierarchical graph clustering.** The most popular cost function for HC is proposed by Dasgupta (2016). Given a weighted graph $G = (V, E, w)$ and a cluster tree $T$, Dasgupta's cost is defined as

$$das\_cost^T(G) = \sum_{i,j \in E} w_{ij} |V(i \vee j)|, \tag{1}$$

where $i \vee j$ denotes the least common ancestor (LCA) of $i$ and $j$ in $T$, and $V(i \vee j)$ represents the set of descendent leaf nodes under $i \vee j$. On similarity-based graphs, optimization of HC trees can be performed by minimizing Dasgupta's objective. The intuition is that for a good clustering tree, the edges with larger weights ought to be placed as far down from the root as possible, which makes the number of leaves covered by its LCA on the HC tree as small as possible. Dasgupta also showed that minimizing $das\_cost^T(G)$ and maximizing its dual $das\_cost^T(G)$ are both NP-hard.

Along this line of study, Dasgupta showed that recursive bipartition applying Arora's seminal algorithm for sparsest cut problem Arora et al. (2009) yields $O(\log^{1.5} n)$-approximation, and it was improved by Roy & Pokutta (2016) and Charikar & Chatziafratis (2017); Cohen-Addad et al. (2019) to $O(\log n)$ and $O(\sqrt{\log n})$, respectively. It is also known to be SSE-hard to achieve any constant approximation factor for this objective Charikar & Chatziafratis (2017). Moseley and Wang studied the dual of Dasgupta's cost function and showed that the average linkage algorithm achieves a $(1/3)$-approximation Moseley & Wang (2017). This factor has been improved by a series of works to $0.336$ Charikar et al. (2019), $0.4246$ Ahmadian et al. (2019) and $0.585$ Alon et al. (2020), respectively. There are also some studies considering the problem of maximizing Dasgupta's cost function on dissimilarity-based graphs Cohen-Addad et al. (2019); Charikar et al. (2019); Rahgoshay & Salavatipour (2021); Naumov et al. (2021).

**Overlapping graph clustering.** Newman and Girvan proposed modularity in 2004 Newman & Girvan (2004), which was one of the most popular cost functions for flat non-overlap clustering. Many researchers have extended modularity to the scope of OC. Nepusz et al. (2008) and Nicosia et al. (2009) proposed the concept of belonging factor, which is used to represent the intensities of a node and an edge belonging to a cluster. A function of the belonging factor was introduced to the definition of modularity to make it applicable to OC, and a heuristic algorithm was proposed based on maximizing OC modularity. Zhang et al. (2007), Shen et al. (2009) and Chen et al. (2010) also proposed their own definitions of belonging factor and cost functions based on modularity. Inspired by these works, our cost function also utilizes belonging factor for HOC.

On the worst-case guarantee analysis for OC, Khandekar et al. (2014) formulated it as the problem that minimizes the maximum or the sum of conductances of overlapping clusters, with or without a bounded number of clusters. They proposed the algorithms that achieve $O(\log n)$-approximation factors for the four kinds of versions, where $n$ is the number of vertices. The techniques behind the proof include the tree decompositions Räcke (2002; 2008); Harrelson et al. (2003) and a dynamic programming. As claimed in their work, the complexity of the dynamic program hinders the scalability of their methods.

Another representative work for OC is attributed to Orecchia et al. (2022), in which two cost functions called $\epsilon$-overlapping ratio-cut ($\epsilon$-ORC) and $\lambda$-hybrid ratio-cut ($\lambda$-HCUT) respectively are proposed for OC with two overlapping clusters. Both cost functions are designed based on the ratio-cut objective, and treat the overlapping part of the two clusters as a penalty. Concretely, given a graph $G = (V, E, w, \mu)$ with non-negative edge weights $w$, vertex measure $\mu$, and two overlapping clusters $L$ and $R$ of vertices, they define two ratio-cut-like measures to be $q_E[L, R] = w(L \setminus R, R \setminus L)/\min\{\mu(L), \mu(R)\}$ and $q_V[L, R] = \mu(L \cap R)/\min\{\mu(L), \mu(R)\}$. Then the $\epsilon$-ORC problem is defined to be the minimization of $q_E[L, R]$ under the condition that $q_V[L, R] \leq \epsilon$, and the $\lambda$-HCUT problem is the minimization of $q_E[L, R] + \lambda q_V[L, R]$. These two problems are defined with hyper-parameters, which restricts the applications and scalability of OC algorithms that solve them. Moreover, since the edge weights $w$ and vertex measure $\mu$ are usually derived from independent systems and have different units, the linear combination of $q_E[L, R]$ and $q_V[L, R]$ in $\lambda$-HCUT is less explainable. However, for both $\epsilon$-ORC and $\lambda$-HCUT, Orecchia et al. (2022) gave a nearly-linear-time $O(\log n)$-approximation algorithms called $cm + improve$ for both of them. $cm + improve$ is scalable to large graphs with tens of millions of edges, and is the main competitor in our experiments.

With regard to HOC, there is much less work. Only a few methods for dissimilarity-based vector data are proposed. Some heuristics based on density criterion Jeantet et al. (2020) and cut metrics Gama et al. (2018) are utilized during the clustering process. But no cost function and theoretical guarantee have been developed yet, which is just what our work addresses.

## 2   A COST FUNCTION FOR HOC

In this section, we formulate our cost function for HOC. First of all, we briefly introduce the underlying idea. HOC can be represented by a directed acyclic graph, called HOC graph, that is a natural generalization of HC tree. Inspired by Dasgupta's cost function for HC, we extend the LCA of an edge to its minimal common ancestor set, and introduce the belonging factor to measure the degree by which a node, a cluster, or an edge belongs to an ancestor. Intuitively, for a similarity-based graph, a quality HOC graph should involve edges of heavy weights into clusters that are small and as far down from the root of the HOC graph as possible. Overlapping is desirable when a node has strong connections to more than one cluster simultaneously, in which case, the belonging factor allows to suppress the cost contributed by the edges incident to the node. This is the crucial idea of our cost function for HOC.

**Preliminaries.** An undirected weighted graph $G = (V, E, w)$ is specified by a node set $V$, an edge set $E \subseteq \{(u, v)|u, v \in V\}$, and a weight function $w : E \to \mathbb{R}^+$. Let $n = |V|$ and $m = |E|$ represent the number of nodes and the number of edges, respectively. The degree of a node $u$, denoted by $d_u$, is the sum of weights of all edges incident to $u$, i.e., $d_u = \sum_{(u,v) \in E} w(u, v)$. The induced subgraph of $G$ on the node set $U$ is denoted by $G[U]$. For any $A, B \subseteq V$, let $E(A) = \{(u, v)|(u, v) \in E, u, v \in A\}$, $E(A, B) = \{(u, v)|(u, v) \in E, u \in A, v \in B\}$, $w(A) = \sum_{(u,v) \in E(A)} w(u, v)$, $w(A, B) = \sum_{(u,v) \in E(A,B)} w(u, v)$. For a node $v \in V, w(v, A) = \sum_{a \in \{a|a \in A, (v,a) \in E\}} w(v, a)$. For any $E_0 \subseteq E, w(E_0) = \sum_{e \in E_0} w(e)$.

Partial ordering relationship of two nodes $N$ and $N'$ on a directed acyclic graph $D$, denoted by $N \leq N'$, means that $N'$ is reachable from $N$, and we say that $N$ and $N'$ are *comparable* in this case, and *incomparable* otherwise. An *anti-chain* $L = \{N_1, N_2, N_3, ...\}$ on $D$ is a set of nodes of $D$ satisfying that any two nodes in $L$ are incomparable. We define the *width* of an HOC graph to be the length of the longest anti-chain that consists of non-leaf nodes. HC on graph $G$ is represented by an HC tree $T$. It has $n$ leaf nodes corresponding to the nodes of $G$. For any internal node $N$ on $T$, let $V(N)$ denote the set of leaf nodes in the subtree that treats $N$ as the root. Let $u \vee v$ denote the LCA of $u$ and $v$ on $T$. A weighted graph $G = (V, E, w)$ is called a *similarity-based* graph if it satisfies that the larger $w(u, v)$ is, the more similar $u$ and $v$ are. The cost function for HOC discussed in this paper is proposed for similarity-based graphs.

**Definition 2.1** (hierarchical overlapping clustering graph). *Given a graph $G$, a hierarchical overlapping clustering graph (HOC graph) $D$ on $G$ is a directed acyclic graph that satisfies the following three constraints: (1) There is only one node of $D$ with out-degree of $0$, referred to as the root node and denoted by $R$. (2) There are $n$ nodes of $D$ with in-degree of $0$, corresponding to all the nodes in $V$, referred to as leaf nodes. (3) For each non-root node of $D$, its parent node set $\{N_1, N_2, ...\}$, which is the collection of nodes it points to directly, forms an anti-chain.*

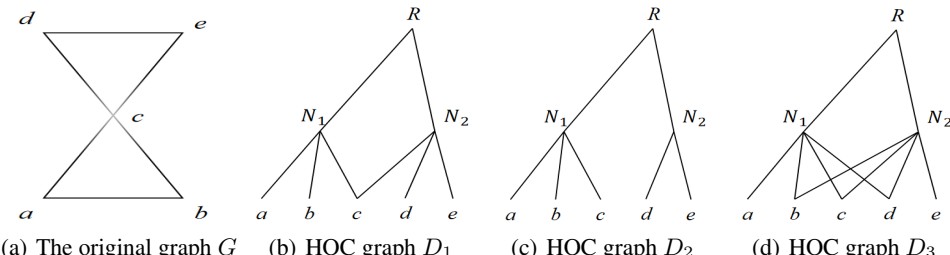

(a) The original graph $G$     (b) HOC graph $D_1$     (c) HOC graph $D_2$     (d) HOC graph $D_3$

Figure 1: An example of HOC graphs.

On an HOC graph, two nodes satisfying $X \leq Y$ means that $V(X)$ is a subset of $V(Y)$. Note that we do not need the converse also holds, because although syntactically we have $V(X) \subseteq V(Y)$, semantically in practice, $X$ and $Y$ may have unrelated meanings from two different systems that are organized by different mechanisms. HOC graph extends the concept of HC tree by allowing each non-root node to have multiple parent nodes that are incomparable with each other. It is a canonical representation for hierarchical set containment that a subset is only allowed to point to a minimal set that contains it. If each non-root node has out-degree one, an HOC graph degenerates into an HC tree. The distance $dis(X, Y)$ is the length of the shortest path from $X$ to $Y$. The height of $D$ is the maximum distance from any leaf to the root, denoted by $h_D = \max_{v \in V} dis(v, R)$. For any node $N$ on $D$, Let $N^-$ denote the set of $N$'s parent nodes and $N_-$ denote the set of $N$'s children nodes. Figure 1 demonstrates three HOC graphs of height 2 for a graph $G$ that consists of two triangles intersecting at a single node, in which $D_2$ is an HC tree without overlap.

**Definition 2.2** (minimal common ancestor set). *The minimal common ancestor set for nodes $u$ and $v$ in $D$ is defined as $M_{uv} = \{N | N \in D, u, v \in V(N), and \forall X \in N_-, u \notin X \text{ or } v \notin X\}$.*

The term "minimal" in the above definition means that any child node of this common ancestor of $u$ and $v$ is not a common ancestor, and thus cannot be further reduced. This is an extension of the unique LCA on HC trees to multiple ones on HOC graphs. For convenience, when $u$, $v$ are two endpoints of an edge, we also say a common ancestor of this edge $(u, v)$. As illustrated in $D_3$ of Figure 1, $M_{bc} = \{N_1, N_2\}$, $M_{ae} = \{R\}$.

Then we introduce belonging factor that is a key ingredient of our cost function. We define two kinds of belonging factors on an HOC graph $D$, node-to-node and edge-to-node belonging factors, that are generalizations of those proposed by Nepusz et al. (2008) and Nicosia et al. (2009) for OC. The belonging factor of node $X$ (resp. edge $(u, v)$) to node $Y$ represents the degree for which $X$ (resp. edge $(u, v)$) belongs to $Y$.

**Definition 2.3** (node-to-node belonging factor). *The node-to-node belonging factor of $X$ to $Y$, denoted by $\alpha_{X,Y}$, is defined recursively. First, define the node-to-node belonging factor for each parent-child node pair on $D$, whose value can be assigned freely as long as it satisfies the following two constraints: (1) $0 \leq \alpha_{X,Y} \leq 1$ for all $X \in D$ and $Y \in X^-$; (2) $\sum_{Y \in X^-} \alpha_{X,Y} = 1$ for each non-root node $X \in D$. Second, for other relationships of $X$ and $Y$, $\alpha_{X,Y}$ is defined as*

$$\alpha_{X,Y} = \begin{cases} \sum_{N \in X^-} \alpha_{X,N} \cdot \alpha_{N,Y} & X \leq Y, X \neq Y \\ 1 & X = Y \\ 0 & otherwise \end{cases} \quad (2)$$

To better understand the belonging factor, it is easy to verify that the above definition is equivalent to the following plain one. For any two comparable nodes $X \leq Y$, denote by $P_{X,Y}$ the set of all paths from $X$ to $Y$. For each path $p = [p_0, p_1, ..., p_{len(p)}] \in P_{X,Y}$, let $p_0 = X$, $p_{len(p)} = Y$, $len(p)$ be the length of $p$. Then the node-to-node belonging factor $\alpha_{X,Y}$ is defined as $\alpha_{X,Y} = \sum_{p \in P_{X,Y}} \prod_{i=0}^{len(p)-1} \alpha_{p_i, p_{i+1}}$ if $X \leq Y, X \neq Y$, and has the same values as Eq. (2) for the other two cases. That is, $\alpha_{X,Y}$ is the sum of the multiplication of all belonging factors of parent-child pairs along each path from $X$ to $Y$. The node-to-node belonging factor has some fundamental properties.

**Property 2.4.** *If $Y$ is the only parent node of $X$, then $\alpha_{X,Y} = 1$.*

**Property 2.5.** *The node-to-node belonging factor of any node to the root is 1, that is, $\alpha_{N,R} = 1$, $\forall N \in D$.*

**Property 2.6.** *For two nodes $X, Y$ of $D$ where $X$ is reachable to $Y$, if there is a node set $S = \{N_1, N_2, ..., N_k\}$ satisfying: (1) $S$ is an anti-chain, (2) $\forall N \in S, X \leq N \leq Y$, (3) $\forall N \in S, \exists p \in P_{X,Y}, N \in p$, (4) $\forall p \in P_{X,Y}, |p \cap S| = 1$. Then $\alpha_{X,Y} = \sum_{N \in S} \alpha_{X,N} \cdot \alpha_{N,Y}$.*

Property 2.4 unifies the HOC graph and the common HC tree. Property 2.5 coincides with the common sense that any cluster and leaf belong totally to the root. Property 2.6 means that the node-to-node belonging factor of $X$ to its ancestor $Y$ can be disassembled by a maximal anti-chain between $X$ and $Y$. The proofs of the above properties are provided in Appendix A.1. Based on node-to-node belonging factor, edge-to-node belonging factor can be defined as follows.

**Definition 2.7** (edge-to-node belonging factor). *For an edge $(u, v)$ in graph $G$, let $X \in M_{uv}$ be one of its minimal common ancestor. The edge-to-node belonging factor $\beta^X_{(u,v)}$ of $(u, v)$ with respect to $X$ is defined as $\beta^X_{(u,v)} = f^X_{(u,v)} / \sum_{Y \in M_{uv}} f^Y_{(u,v)}$, where $f^X_{(u,v)} = \alpha_{u,X} \cdot \alpha_{v,X}$.*

$\beta^X_{(u,v)}$ is normalized over all minimal common ancestors of $(u, v)$ to guarantee that the mass of its belonging factors sums up to 1 over all clusters that $(u, v)$ belongs to. A natural option is the uniform allocation to each parent. Formally, for a node $X \in D$,

$$\alpha_{X,Y} = \begin{cases} \frac{1}{|X^-|} & Y \in X^- \\ 0 & Y \notin X^- \end{cases} \tag{3}$$

We adapt this definition of belonging factor in Section 3. As illustrated in $D_1$ of Figure 1, leaves $c$ has two minimal ancestors $N_1$ and $N_2$, for each of which has belonging factor $1/2$, and all edges in $G$ belongs totally to $N_1$ or $N_2$. In $D_3$, leaves $b$, $c$ and $d$ have both $N_1$ and $N_2$ as their minimal ancestors with belonging factor $1/2$ to each, and the edge-to-node belonging factors of $(b, c)$ and $(c, d)$ to either $N_1$ or $N_2$ are $1/2$. We also demonstrate another toy example in Appendix A.2.

Now, we are ready to introduce our HOC cost function based on the edge-to-node belonging factor.

**Definition 2.8** (cost function for HOC). *Given a graph $G$ and an HOC graph $D$ of $G$, the cost function of $D$ on $G$ is defined as*

$$H^D(G) = \sum_{(u,v) \in E} \left( w(u,v) \cdot \sum_{N \in M_{uv}} \beta^N_{(u,v)} \cdot |V(N)| \right).$$

The cost function contains two summations. The first is over all edges, and the second is over the minimal common ancestors of the endpoints of corresponding edge. The cost contributed by each edge is given by $w(u,v) \sum_{N \in M_{uv}} \beta^N_{(u,v)} \cdot |V(N)|$. Compared with Dasgupta's cost function (1), $H^D(G)$ generalizes it from HC to HOC by assigning a belonging factor for each minimal common ancestor of each edge.

**Definition 2.9** (HOC problem). *The HOC problem on a similarity-based graph $G$ is defined as $\min_D H^D(G)$ under some proper constraints on the HOC graph $D$.*

The intuition behind minimizing the cost function on similarity-based graphs is the same as Dasgupta's cost function $das\_cost$, that is, to assign heavy edges to the clusters as small as possible. On an HOC graph, this can be achieved by ensuring that the minimal common ancestors of these edges are as far down from the root as possible.

As illustrated in Figure 1, according to Definition 2.8, the costs of $D_1$, $D_2$ and $D_3$ are 18, 21 and 24, respectively. We provide the calculating process in Appendix A.3. We can see that $D_1$ has the smallest cost, which indicates that $D_1$ is a more reasonable overlapping clustering graph than $D_2$ and $D_3$. Obviously, $D_1$ is more consistent with our intuition. This instance also demonstrates that introducing overlaps has the advantage of reducing the minimal common ancestors of edges, thereby decreasing their costs (compare $D_2$ to $D_1$). On the other hand, this comes at the expense of increasing the number of descendant leaf nodes of the ancestors. So, excessive overlap gets punished (compare $D_1$ to $D_3$). Therefore, our cost function balances the two cases of non-overlap and excessive overlap.

**Remark.** Note that HOC is quite different from HC since it allows possibly an exponential number of overlapping clusters without any restriction, and thus proper constraints on $D$ are necessary. However, we need to be very careful in formulating the constraints. In fact, there is a trivial solution that allows

two endpoints of each edge to form a cluster, which achieves the minimum cost $2w(E)$. Treating two endpoints of each edge as a cluster is in fact an intuitive way for overlapping cluster settings, but due to the large number of clusters, it is meaningless. This is quite different from the optimization of Dasgupta's cost function for HC. A natural restriction on HOC graphs may be on the number of clusters. However, since an HOC graph has hierarchical clusters, we seek to have a meaningful constraint on the cluster number. To this end, we utilize the width of an HOC graph, which is the longest anti-chain on it.

**Definition 2.10** ($k$-HOC problem). *The $k$-HOC problem on a similarity-based graph $G$ is defined as* $\min_D H^D(G)$ *for which the width of the HOC graph $D$ is at most $k$.*

To better understand this problem, let us consider a non-overlapping HC tree first. Here, the width means the largest number of bottom and smallest non-overlapping clusters that contain the leaves directly. These can be considered as a set of key clusters that are closest to the leaves on the tree. Similarly, on an HOC graph, since the longest anti-chain blockades all paths from leaves to the root, the width measures intuitively the number of the incomparable key clusters that contain the leaves.

Moreover, we define $k$-*OC problem* to be the $k$-HOC problem in which we additionally restrict the height of $D$ at most 2, in which case HOC degrades to OC. A fundamental case is 2-OC that allows only two overlapping clusters. 2-OC can be considered as a key ingredient of HOC with multiple clusters since it could be a nice way to construct a $k$-HOC graph by recursively calling 2-OC algorithm in a top-down fashion. In Section 3, our algorithm for $k$-HOC proceeds in this way.

Next, we give some fundamental properties of our HOC cost function, and prove them in Appendix A.4.

**Property 2.11** (compatibility). *If $D$ is restricted to be an HC tree, then $H^D(G) = das\_cost^D(G) = \sum_{(u,v)\in E} w(u,v) \cdot |u \vee v|$.*

**Property 2.12** (additivity on nodes). *For any node $N$ on $D$, let $E_N^D$ denote the set of edges with $N$ as a minimal common ancestor, i.e., $E_N^D = \{(u,v)|(u,v) \in E, N \in M_{uv}\}$. The HOC cost function can be rewritten as: $H^D(G) = \sum_{N \in D} \left(|V(N)| \cdot \sum_{(u,v)\in E_N^D} w(u,v)\beta_{(u,v)}^N\right)$.*

**Property 2.13** (binary optimality). *When the number of nodes of $D$ is unbounded, there is an optimal HOC graph that is binary, i.e., the number of children of each node is at most 2.*

Property 2.11 indicates that our cost function for HOC can be unified with Dasgupta's cost. That is, under the constraint of hierarchical non-overlapping clustering, our cost function for HOC problem degrades to Dasgupta's objective whose optimization is NP-hard Dasgupta (2016). Property 2.12 provides an alternative interpretation of the cost function from another perspective, for which it can be seen as the sum of costs associated with each node. Property 2.13 describes the structure of the optimal HOC graph with unbounded number of nodes, and Dasgupta's cost also has this property.

**Primal and dual versions of HOC problem.** Next, we introduce the primal and the dual versions of the HOC problem. Note that besides the trivial lower bound $2w(E)$ for $\min_D H^D(G)$, we also have a trivial upper bound $nw(E)$, since the size of any common ancestor of two leaves on $D$ is at most $n$. So, we define the primal HOC problem, denoted by $k$-HOC-P, to be $\min_D H^D(G)$ as we have defined in Definition 2.9. We define the dual HOC problem, denoted by $k$-HOC-D, to be $\max_D\{nw(E) - H^D(G)\}$, where by Definition 2.8, $nw(E) - H^D(G) = \sum_{(u,v)\in E} \left(w(u,v) \cdot \sum_{N \in M_{uv}} \beta_{(u,v)}^N \cdot (n - |V(N)|)\right)$. The solutions to primal and dual problems achieve optima on the same HOC graph. Similarly, $k$-OC-P and $k$-OC-D denote the corresponding version of OC problem, respectively.

## 3 An algorithm for $k$-HOC

In this section, we propose our algorithm for the $k$-HOC problem. We use the Equation (3) as the node-to-node belonging factor. As mentioned earlier, we first study the fundamental case of 2-OC, and then apply it to $k$-HOC. The 2-OC problem has its own interests.

## 3.1 AN APPROXIMATION ALGORITHM FOR 2-OC

**Cost functions for 2-OC.** In the 2-OC setting, given a graph $G = (V, E, w)$, we restrict the height of the HOC graph to 2 and the number of children of the root $R$ to 2. Suppose that two clusters $N_1 = A \cup B$ and $N_2 = C \cup B$ overlap on $B$. Definition 2.8, the cost function of 2-OC-P can be formulated as $cost_{primal}(A, B, C) = [w(A) + w(A, B)](|A| + |B|) + [w(B, C) + w(C)](|B| + |C|) + \frac{(|A| + 2|B| + |C|)w(B)}{2} + w(A, C)n$, and 2-OC-P can be formulated as

$$\min_{A,B,C \subseteq V} cost_{primal}(A, B, C) \tag{2-OC-P}$$

We also have the cost function $cost_{dual}(A, B, C) = (w(A+B) - \frac{w(B)}{2})|C| + (w(B+C) - \frac{w(B)}{2})|A|$ for 2-OC-D, and 2-OC-D can be formulated as

$$\max_{A,B,C \subseteq V} cost_{dual}(A, B, C) \tag{2-OC-D}$$

The derivation processes of the forms of $cost_{primal}$ and $cost_{dual}$ are presented in Appendix B.1. We remark that although our cost functions for 2-OC look complicated, they are hyper-parameter free and natural from the perspective of HOC, which is superior to the objective proposed by Orecchia et al. (2022). Then we propose our algorithm for 2-OC.

---

**Algorithm 1:** Algorithm for 2-OC

---

**Input:** an undirected graph $G = (V, E, w)$
**Output:** node sets $A$, $B$ and $C$ for 2-OC
1 $n \leftarrow |V|, p \leftarrow \frac{1}{\sqrt{6}}, x \leftarrow \frac{p}{1+2p}$;
2 Define a new $cost_{temp}(A, B, C) = w(E) - w(A, C) + xw(B)$;
3 Divide arbitrarily $V$ into three disjoint parts $A, B, C$ satisfying $|A| = |C| = pn$, $|B| = (1 - 2p)n$, such that two endpoints of the edge with the largest weight are both in $A$;
4 **repeat**
5     Exchange any two nodes from different sets of $A, B, C$ whenever $cost_{temp}$ can be amplified by more than $1 + 1/\epsilon n^2$ times;
6 **until** *get stuck*;
7 return $A, B, C$.

---

**Approximation algorithm for 2-OC.** Algorithm 1 is a simple local search process for 2-OC. It first defines a surrogate cost function $cost_{temp}(A, B, C) = w(E) - w(A, C) + xw(B)$, and initializes $A$, $B$, $C$ arbitrarily (e.g. a random initialization). After that, the nodes in $A$, $B$, $C$ exchange pairwisely on the condition that current cost can be amplified by $1 + 1/\epsilon n^2$ times, that is, $cost_{temp}(A', B', C') > (1 + \frac{\epsilon}{n^2})cost_{temp}(A, B, C)$, where $A'$, $B'$, $C'$ are the node sets after exchanging corresponding to $A, B, C$ respectively. It doesn't terminate until no pair of nodes meets the exchange condition. For the worst-case guarantee, we have the following theorem.

**Theorem 3.1.** *Algorithm 1 achieves an approximation factor $a = \frac{2}{3\sqrt{6}} - \Theta(\frac{1+\epsilon}{n})$ for 2-OC-D with time complexity $O(\epsilon^{-1}n^4 \log m)$ for any $\epsilon > 0$.*

The idea of the proof of Theorem 3.1 is as follows. Since $nw(E)$ is a trivial upper bound on the objective function, we only have to show that $cost_{dual} \geq \left(\frac{2}{3\sqrt{6}} - \Theta(\frac{1+\epsilon}{n})\right) \cdot nw(E)$. Since Algorithm 1 fixes the sizes of $A$, $B$, $C$, we only need to build the relationship between $w(E)$ and edge weights of different parts in the objective function. A lower bound on the latter related to $w(E)$ (Inequality (11)) can be obtained by the three stuck exchange conditions when the iteration terminates. The detailed proof of Theorem 3.1 is provided in Appendix B.2. Moreover, we have the following proposition to demonstrate the tightness of our guarantee in some sense.

**Proposition 3.2.** *There is an instance $I$ whose optimal value $OPT(I) = \left(\frac{2}{3\sqrt{6}} - \Theta(\frac{1}{n})\right) nw(E)$.*

Proposition 3.2 implies that, if an approximation algorithm for 2-OC-D is designed based on the upper bound $nw(E)$ of $cost_{dual}$, the optimal approximation ratio cannot be better than $\frac{2}{3\sqrt{6}} - \Theta(\frac{1}{n})$. The detailed proof of Proposition 3.2 is provided in Appendix B.3.

In Appendix B.4, we show that Algorithm 1 is also a good approximation algorithm for 2-OC-P.

## 3.2 An approximation algorithm for $k$-HOC

Now we turn to the $k$-HOC problem. We assume that $k \leq n$ for practical significance. Since the width of the HOC graph is no more than $k$, we invoke the 2-OC algorithm $k-1$ times to guarantee this. We first construct a binary tree (excluding the leaves) for the internal nodes, and then merge the identical ones that consists of the same set of leaves, while keeping all directed edges on them. In each iteration, the splitting cluster is chosen greedily according to the relative benefit of cost. Formally, we define $\Delta(X) = \frac{cost_{dual}(X)}{|X|w(X)}$ for the most bottom clusters $X$, where $cost_{dual}(X)$ is the dual cost obtained by the 2-OC algorithm on the subgraph induced by $X$. In each round, we choose the $X$ with the largest $\Delta(X)$ to split. This procedure is described as Algorithm 2.

---

**Algorithm 2:** Algorithm for $k$-HOC

**Input:** an undirected graph $G = (V, E, w)$, an integer $k \leq n$
**Output:** a $k$-HOC graph $D$
1 initialize $D$ with all leaves pointing to the root $r$;
2 $\mathcal{S} \leftarrow \{r\}$;
3 **repeat**
4      $X_{max} \leftarrow \arg\max_{X \in \mathcal{S}}\{\Delta(X)\}$;
5      Apply Algorithm 1 to the subgraph induced by $X_{max}$ and obtain two internal nodes $X_L$, $X_R$;
6      $\mathcal{S} = \mathcal{S} \setminus \{X_{max}\}$;
7      $\mathcal{S} = \mathcal{S} \cup \{X_L, X_R\}$;
8      Add $X_L and X_R$ to $D$ as $X_{max}$'s left and right child, respectively, and redirect the leaves to their corresponding parents;
9 **until** $|\mathcal{S}| = k$;
10 Merge identical nodes in $D$ into a single one while keeping all the connections on them;
11 Remove all redundant directed edges $(X, Y)$ for which there is another path from $X$ to $Y$ in $D$;
12 return $\mathcal{D}$.

---

Now we show that $D$ output by Algorithm 2 is a legal $k$-HOC graph. By definition 2.1, we have to show that the parents of any non-root node form an anti-chain, and the width of $D$ is at most $k$. For any node $X$, since we remove all the directed edges $(X, Y)$ for which there is another path from $X$ to $Y$ in $D$, $X^-$ is obviously an anti-chain. Since the 2-OC algorithm is called for at most $k-1$ times, the width of $D$ before merging is at most $k$. Since merging does not increase the width, the final $D$ is a $k$-HOC graph. For the approximation guarantee, we have the following theorem.

**Theorem 3.3.** *Algorithm 2 achieves an approximation factor $\frac{2}{3\sqrt{6}} - \Theta(\frac{1+\epsilon}{n})$ for the dual version of the $k$-HOC problem.*

To prove Theorem 3.3, we only have to show that the dual cost is at least that of Algorithm 1, which is lower bounded by $\left(\frac{2}{3\sqrt{6}} - \Theta(\frac{1+\epsilon}{n})\right) \cdot nw(E)$. Then the approximation factor follows from the fact that $nw(E)$ upper bounds the dual cost of any HOC graph. We prove it formally in Appendix B.6.

**Time complexity.** The runtime of Algorithm 2 consists of three parts: the recursive division, merging identical nodes and removing redundant edges. In the division step, it calls Algorithm 1 $k$ times, which takes $O(k\epsilon^{-1}n^4 \log m)$ time. In the node merging step, an efficient way of implementation leverages bitmaps and sorting. The bitmap of each internal node indicates the membership of each leaf, and its length is $n$. It is necessary to check whether $O(k)$ bitmaps are the same, which takes $O(nk \log k)$ time. In the edge removing step, a redundant edge $(X, Y)$ can be decided by reversing it and checking whether there is a cycle containing $X$ and $Y$. This takes $O((n+k)^2)$ time. Combining the above three parts and noting that $k \leq n$, the total runtime is $O(k\epsilon^{-1}n^4 \log m)$.

**A speed-up version.** Algorithms 2 and 1 have theoretical significance, but are not efficient enough in practice. Moreover, the setting of fixed sizes of $A$, $B$ and $C$ in Algorithms 1 is too rigid to fit for flexible scenarios. For scalability and practical application of our algorithm, we propose the speed-up version (Algorithm 3) of Algorithm 1 and use it in Algorithm 2 to yield our speed-up algorithm for $k$-HOC. Their effectiveness and scalability will be verified in Section 4. Two easy heuristics are proposed for speed-up, and the pseudocode of Algorithm 3 is presented in Appendix B.7.

(1) Initialization based on ratio-cut Hagen & Kahng (1992): Instead of the random strategy for the initial trisection, we use the spectral clustering algorithm RatioCut to split the node set into two pieces, denoted by $X, Y$, let $A = X$, $B = \emptyset$, $C = Y$. Then the nodes move greedily among $A$, $B$, and $C$ instead of exchange.

(2) Batch migration: Starting from the initial $A$, $B$, $C$, calculate the variation of the cost for each node when it moves to another set, and select a batch of $\gamma|V|$ nodes (if any) with positive and the largest variation of cost to move in one step UNTIL all nodes get stuck. In our experiment, we set $\gamma = 0.02$. If this threshold is not reached, we just move all nodes that need to move.

## 4 EXPERIMENTS

In this section, we verify by experiments the effectiveness and scalability of the speed-up version of Algorithm 2, which also demonstrates the validity of our cost function as well. All experiments were performed on a computer equipped with a 2.3GHz quad-core Intel i5 processor with 8GB memory. For the source codes and datasets, please refer to the supplementary materials.

**Baseline.** We include two baseline methods. The first one is OHC'20 proposed by Jeantet et al. (2020), which is a density-based algorithm for HOC in an agglomerative bottom-up fashion. It works only for dissimilarity-based vector data. To fit to graph clustering in our experiments, we feed to OHC'20 as input the spectral embedding consisting of the top-$k$ eigenvectors of the Laplacian matrix. Since this method need to deal with all-pair distance, it cannot work on large graphs. The second one is $cm + improve$ proposed by Orecchia et al. (2022), which is a nearly linear-time overlapping bipartition algorithm with $O(\log n)$-approximation. However, due to the version issue and complicate organization of the source files, we are not able to compile correctly their codes published online. So we compare our 2-OC algorithm with $cm + improve$ on the same datasets as Orecchia et al. (2022) uses by moving their results to our table directly (Table 1, the last column).

**Synthetic datasets.** We use the overlapping stochastic block model (OSBM). OSBM is a generalization of SBM such that the preset $k$ clusters overlap. We modify it to preset two hierarchies by setting the first level inter-link probability $p_1$, the second level inter-link probability $p_2$, and the intra-link probability $p_3$. We give its definition in Appendix C.1. We use NMI for OC Lancichinetti et al. (2009) to evaluate our algorithm, and its formal definition is provided in Appendix C.2. Since such NMI is only fit to non-hierarchical clusters, we evaluate our algorithm results on each level of HOC graph.

**Real datasets.** For a fair comparison, we adopt the real datasets on `http://snap.stanford.edu/data` including Amazon, Youtube, and DBLP Yang & Leskovec (2012b) that are also used by $cm + improve$ Orecchia et al. (2022). Because of lacking ground truth for HOC, we only evaluate scalability on the real datasets.

**Effectiveness.** We demonstrate in Figure 2 the results on OSBM datasets with varying sizes. We show the time, cost, and NMI of our $k$-HOC algorithm, and compare it with OHC'20, as well as the non-overlapping version that sets $B$ in Algorithm 1 empty and thus degrades to optimizing Dasgupta's cost. It can be observed that the runtime of our HOC algorithm that generates four overlapping bottom clusters for dense graph of size 5000 is only around 80s, and that for sparse graph of size $10^5$ is less than 15min. We do not show the results of OHC'20 for sparse graphs since it is not able to terminate in one hour for a graph of size $10^4$. The cost results indicate that our algorithm outperforms OHC'20, and we have indeed gained benefits of cost from overlapping when compared with the non-overlapping counterpart of Dasgupta's cost. We evaluate NMI on the two hierarchies respectively. For OHC'20, since it cannot restrict the hierarchy numbers, in each round of evaluation, we choose the level that achieves the highest NMI compared with the ground truth. Most NMIs are above 0.9, which demonstrates that our $k$-HOC algorithm achieves high accuracy in reconstructing hierarchical overlapping clusters on each level. We also visualize a result in Appendix C.4.

**Scalability.** Figure 2 has demonstrated that our $k$-HOC algorithm has good scalability in synthetic graphs. Next, we show in Table 1 the scalability of our algorithm for 2-OC on large real datasets. It can be seen that the runtime of our algorithm on all the datasets is much shorter than that of the

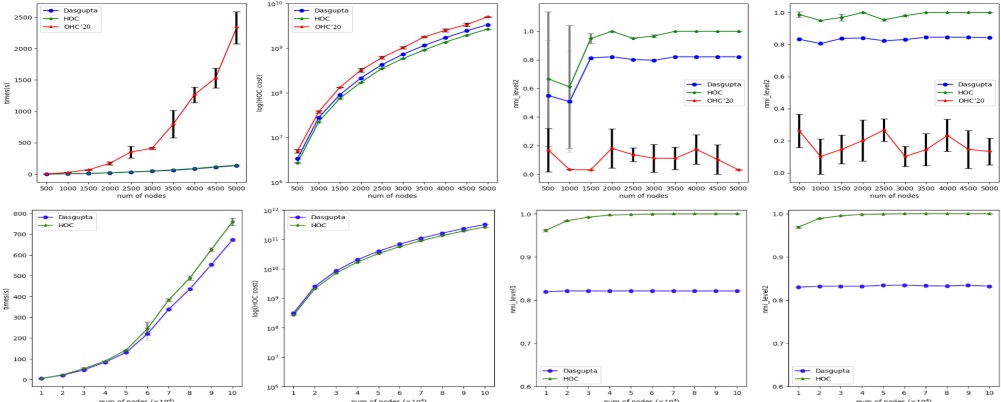

Figure 2: The results of time, cost, and NMI. In each figure, the $x$-axis indicates the graph size. The graphs in the first row are small and dense, while those in the second row are large and sparse. We take $k = 4$ in OSBM, and in each cluster, the size ratio of overlapping to non-overlapping is 9:1. In the fist row, $p_1 = 10^{-3}$, $p_2 = 5 \times 10^{-3}$, $p_3 = 0.5$. In the second row, $p_1 = 10^{-4}$, $p_2 = 2 \times 10^{-4}$, $p_3 = 5 \times 10^{-3}$. Regard to the last two columns of NMI results, "level 1" is the first level that contains the two high-level clusters, and "level 2" is the second one that contains the four low-level clusters. Each point is calculated on average over 5 trials, and error bar indicates standard deviation.

baseline method $cm + improve$ [1]. Especially, on Youtube dataset that has around one million nodes and three million edges, the runtime of our speed-up algorithm implemented on a single personal computer is less than 12 minutes, which is only around 20% of the runtime of $cm + improve$ that is run on a server. Although Orecchia et al. (2022) showed that $cm + improve$ has nearly linear runtime, which is built on the recent solid work Chen et al. (2022) that has provided a nearly linear-time algorithm for the maximum-flow problem, they actually used the HIPR implementation Cherkassky et al. (1994) with the push-labeled method for this. The advantage of our algorithm in efficiency benefits from our new cost function and the simple local search strategy.

Table 1: Scalability performance on real datasets

| dataset | n | m | time | cm time |
| --- | --- | --- | --- | --- |
| Amazon | 334863 | 925872 | <3min | 15-18min |
| Youtube | 1134890 | 2987624 | <12min | 55-75min |
| DBLP-all | 317080 | 1049866 | <3min | – |
| DBLP-cm | 83114 | 409541 | <21s | 2-4min |

## 5  CONCLUSIONS AND FUTURE WORK

**Conclusions.** In this paper, we study the problem of hierarchical overlapping clustering from the aspects of cost function, algorithm and scalability. We propose a cost function and give some basic properties. We provide an approximation algorithm that achieves constant factor for the dual version of $k$-HOC problem. A speed-up version of our algorithm based on some easy heuristics during local search has good performances in HOC reconstruction and good scalability.

**Future work.** There are many directions worth further study. The first is about approximation algorithm for the primal $k$-HOC problem for $k > 2$. Although we know the complementary relationship between the primal and the dual problems, the approximation guarantees are quite different. The second is about variant versions of the HOC problem, e.g., having other constraints on HOC graphs and alternative definitions of node-to-node and edge-to-node belonging factors. These flexible settings may adapt to different application scenarios.

---

[1]The results in the last column of Table 1 are from Table 3 of the original paper Orecchia et al. (2022) whose experimental operating environment includes a cluster of machines with 24 Cores (2x 24 core Intel Xeon Silver 4116 CPU @ 2.10GHz), 48 threads and 128GB RAM. In contrast, we have only used a personal computer.

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

## A  SUPPLEMENT TO THE COST FUNCTION FOR HOC

In this section, we provide some supplements for our cost function for HOC.

### A.1  PROOFS OF THE PROPERTIES OF BELONGING FACTOR

(1) Proof of Property 2.4

*Proof.* By the definition of $\alpha$, $\sum_{Y \in X^-} \alpha_{X,Y} = 1$. □

(2) Proof of Property 2.5

*Proof.* We prove it by induction. We group the nodes on an HOC graph by the distances from the root $R$. Let $P = \{L_1, L_2, ...\}$, where $L_i = \{v | dis(v, R) = i\}$. Then we prove the property by induction on $i$. $\forall N \in L_1$, since $|N^-| = 1$, we have $\alpha_{N,R} = 1$. Suppose that $\forall X \in L_k$, $\alpha_{X,R} = 1$, then $\forall N \in L_{k+1}$, by the recursive definition of $\alpha$, $\alpha_{N,R} = \sum_{X \in N^-} \alpha_{N,X} \cdot \alpha_{X,R} = \sum_{X \in N^-} \alpha_{N,X} = 1$. □

(3) Proof of Property 2.6

*Proof.* It can be verified directly by the definition of $\alpha$.

$$\alpha_{X,Y} = \sum_{p \in P_{X,Y}} \prod_{i=0}^{len(p)-1} \alpha_{p_i,p_{i+1}}$$

$$= \sum_{N \in S} \sum_{p:|p \cup N|=1} \prod_{i=0}^{len(p)-1} \alpha_{p_i,p_{i+1}}$$

$$= \sum_{N \in S} \left( \sum_{p \in P_{X,N}} \prod_{i=0}^{len(p)-1} \alpha_{p_i,p_{i+1}} \right) \left( \sum_{p \in P_{N,Y}} \prod_{i=0}^{len(p)-1} \alpha_{p_i,p_{i+1}} \right)$$

$$= \sum_{N \in S} \alpha_{X,N} \cdot \alpha_{N,Y}$$

□

### A.2  A TOY EXAMPLE OF BELONGING FACTOR

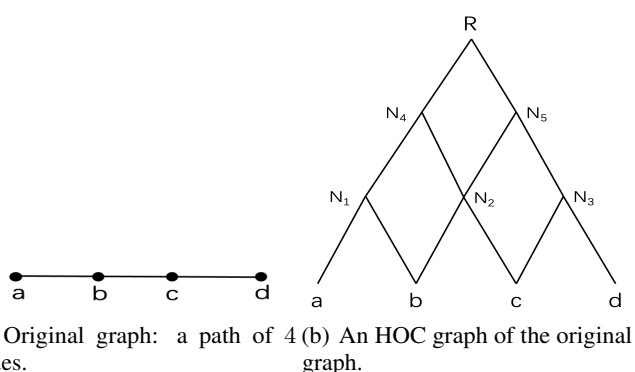

(a) Original graph: a path of 4 nodes.

(b) An HOC graph of the original graph.

Figure 3: An illustration of HOC graph.

In order to better understand node-to-node and edge-to-node belonging factors, we give an example in this section.

As shown in Figure 3, graph $G$ is a path of 4 nodes, and a possible HOC graph is shown in Figure (b).

Table 2 demonstrates the minimal common ancestor set of each leaf node pair. Table 3 shows the node-to-node belonging factor of each child-to-parent node pair on the HOC graph, and those of any others can be calculated by Definition 2.3. For example,

$$\alpha_{b,N_4} = \alpha_{b,N_1} \cdot \alpha_{N_1,N_4} + \alpha_{b,N_2} \cdot \alpha_{N_2,N_4} = \frac{1}{2} \times 1 + \frac{1}{2} \times \frac{1}{2} = \frac{3}{4},$$

$$\alpha_{b,N_5} = \alpha_{b,N_2} \cdot \alpha_{N_2,N_5} = \frac{1}{2} \times \frac{1}{2} = \frac{1}{4}.$$

Table 2: Minimal common ancestor set

| node pair $(u,v)$ | $(a,b)$ | $(a,c)$ | $(a,d)$ | $(b,c)$ | $(b,d)$ | $(c,d)$ |
|---|---|---|---|---|---|---|
| $M_{uv}$ | $\{N_1\}$ | $\{N_2\}$ | $\{R\}$ | $\{N_2\}$ | $\{N_5\}$ | $\{N_3\}$ |

Table 3: node-to-node belonging factor (child to parents)

| node pair $(X,Y)$ | $(a,N_1)$ | $(b,N_1)$ | $(b,N_2)$ | $(c,N_2)$ | $(c,N_3)$ | $(d,N_3)$ |
|---|---|---|---|---|---|---|
| $\alpha_{X,Y}$ | 1 | $1/2$ | $1/2$ | $1/2$ | $1/2$ | 1 |
| node pair $(X,Y)$ | $(N_1,N_4)$ | $(N_2,N_4)$ | $(N_2,N_5)$ | $(N_3,N_5)$ | $(N_4,R)$ | $(N_5,R)$ |
| $\alpha_{X,Y}$ | 1 | $1/2$ | $1/2$ | 1 | 1 | 1 |

We can also verify the properties of the node-to-node belonging factor. Here we only verify Property 2.6, and other properties can be easily verified. Let $X = b, Y = N_4, S = \{N_1, N_2\}$. We can verify that $S$ satisfies all conditions of Property 2.6. Then

$$\alpha_{b,N_4} = \sum_{N \in S} \alpha_{b,N} * \alpha_{N,N_4} = \alpha_{b,N_1} \cdot \alpha_{N_1,N_4} + \alpha_{b,N_2} \cdot \alpha_{N_2,N_4} = \frac{3}{4}$$

Table 4: edge-to-node belonging factor

| edge $(u,v)$ | minimum common ancestor $N$ | $\beta^N_{(u,v)}$ |
|---|---|---|
| $(a,b)$ | $N_1$ | 1 |
| $(b,c)$ | $N_2$ | 1 |
| $(c,d)$ | $N_3$ | 1 |

Table 4 shows edge-to-node belonging factors of all edges and their minimum common ancestors. Because every edge has only one minimum common ancestor, the edge-to-node belonging factor is 1.

### A.3 COST CALCULATION FOR THE RUNNING EXAMPLE

For reading convenience, we demonstrate the example again.

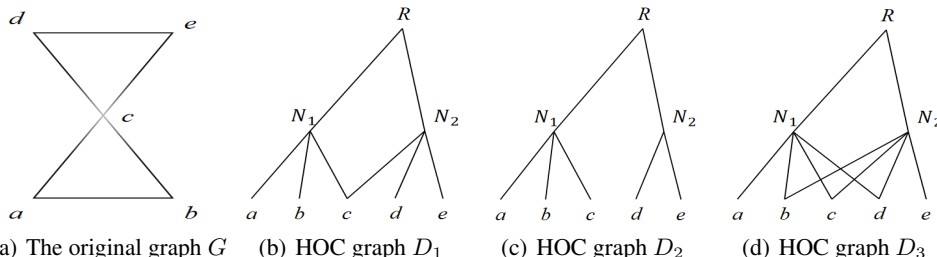

(a) The original graph $G$    (b) HOC graph $D_1$    (c) HOC graph $D_2$    (d) HOC graph $D_3$

Figure 4: An example of HOC graphs.

In $D_1$, all edges have only one minimal common ancestor, so the edge-to-node belonging factors of them are 1. The graph contains 6 edges in all, and each minimal common ancestor has 3 descendant leaf nodes, resulting in the cost $H^{D_1}(G) = 6 \times 3 = 18$.

$D_2$ is not overlapping. For $(a, b)$, $(a, c)$, $(b, c)$, their minimal common ancestor has 3 descendant leaf nodes. For $(d, e)$, the minimal common ancestor has 2 descendant leaf nodes. For $(c, d)$, $(c, e)$, their minimal common ancestor has 5 descendant leaf nodes. All together, the cost $H^{D_2}(G) = 3 \times 3 + 2 + 2 \times 5 = 21$.

In $D_3$, consider 6 terms separately corresponding to the 6 edges. Taking $(b, c)$ as an example, it has two minimal common ancestors. Due to symmetry, the edge-to-node belonging factors of $(b, c)$ regarding to both ancestors are 0.5. Therefore, the cost contributed by $(b, c)$ is $0.5 \times 4 + 0.5 \times 4 = 4$. Thus, the cost $H^{D_3}(G) = 1 \times 4 + 1 \times 4 + (0.5 \times 4 + 0.5 \times 4) + (0.5 \times 4 + 0.5 \times 4) + 1 \times 4 + 1 \times 4 = 24$.

### A.4 PROOFS OF PROPERTIES OF THE COST FUNCTION

(1) proof of Property 2.11 (compatibility)

*Proof.* When $D$ is an HC tree, the minimal common ancestor of edge $(u, v)$ is unique and degenerates to the LCA on the HC tree, and the edge-to-node belonging factor is also 1. Then, we get

$$
\begin{aligned}
H^D(G) &= \sum_{(u,v) \in E} \left( w(u,v) \sum_{N \in M_{uv}} \beta^N_{(u,v)} \cdot |V(N)| \right) \\
&= \sum_{(u,v) \in E} w(u,v) \cdot |u \vee v| \\
&= das\_cost^D(G)
\end{aligned}
$$

$\square$

(2) Proof of Property 2.12 (additivity on nodes)

*Proof.*

$$
\begin{aligned}
H^D(G) &= \sum_{(u,v) \in E} \left( w(u,v) \sum_{N \in M_{uv}} \beta^N_{(u,v)} \cdot |V(N)| \right) \\
&= \sum_{N \in D} \left( |V(N)| \cdot \sum_{(u,v) \in E^D_N} w(u,v) \beta^N_{(u,v)} \right)
\end{aligned}
$$

$\square$

(3) Proof of Property 2.13 (binary optimality)

*Proof.* As shown in Figure 5, assume that (a) represents a local optimum of the optimal solution $D$, where node $N$ has three children: $N_1$, $N_2$, and $N_3$. For any edge $(u, v)$ treating $N$ as a minimal common ancestor, $u$ and $v$ cannot belong to any single cluster of $N_1$, $N_2$, and $N_3$ simultaneously, since otherwise, $N$ would not be the minimal common ancestor for them. Without loss of generality, let's assume that $u$ belongs to $N_1$ and $v$ belongs to $N_2$ (or $N_2$ and $N_3$).

Now, we construct a new node $X$ as the parent of $N_1$ and $N_2$, resulting in the transformed structure shown in (b). As a result, the minimal common ancestor for $(u, v)$ becomes $X$. We observe the following.

Since every path from $u$ to $N'$ passes through $X$ and $\alpha_{X,N'} = 1$, we have $\alpha_{u,N} = \alpha_{u,N'} = \alpha_{u,X}$. Since a path from $v$ to $N$ corresponds to a path from $v$ to $X$, but $v$ may also belong to $N_3$, we have $\alpha_{v,N} \geq \alpha_{v,X}$. Therefore, $\beta^N(u,v) \geq \beta^X(u,v)$, indicating that the edge-to-node belonging factor of $(u, v)$ to $X$ is less than or equal to its edge-to-node belonging factor to $N$. Additionally, we have $|X| < |V(N)|$, implying that

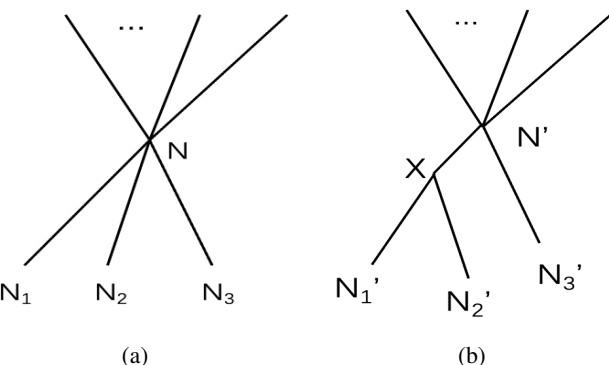

Figure 5: Binary property proof graph

$w(u,v)\beta^X(u,v)|X| < w(u,v)\beta^N(u,v)|V(N)|$. This leads to a reduction in the cost function associated with this term. It is important to note that the cost reduction holds true for any edge treating $N$ as a minimal common ancestor. Moreover, by Property 2.12, the cost function can be expressed as $\sum_{N \in D} |V(N)| \cdot \sum_{(u,v) \in E_N^D} w(u,v)\beta^N_{(u,v)}$. The above operation affects only the cost of a single node. For other nodes, the edge-to-node belonging factor and the number of descendant leaf nodes remain unchanged, thus their values do not change. As a result, the overall cost function decreases. In this way, we can transform any case where a node has more than two children into a binary structure, resulting in a lower cost function value. Therefore, when the number of nodes of $D$ is unbounded, the binary HOC graph constructed above is optimal. $\qquad\square$

## B SUPPLEMENT TO THE ALGORITHMS

In this section, we provide some supplements to our algorithms.

### B.1 PRIMAL AND DUAL PROBLEMS OF 2-OC

Following is the derivation process of the forms of $cost_{primal}$ and $cost_{dual}$ for 2-OC.

$$
\begin{aligned}
\min_D H^D(G) &= \sum_{(u,v) \in E} w(u,v) \sum_{N \in M_{uv}} \beta^N_{(u,v)} \cdot |V(N)| \\
&= \sum_{(u,v) \in E(A)} w(u,v)(|A|+|B|) + \sum_{(u,v) \in E(A,B)} w(u,v)(|A|+|B|) \\
&\quad + \sum_{(u,v) \in E(C)} w(u,v)(|B|+|C|) + \sum_{(u,v) \in E(B,C)} w(u,v)(|B|+|C|) \\
&\quad + \sum_{(u,v) \in E(B)} w(u,v)\left(\beta^{N_1}_{(u,v)} \cdot (|A|+|B|) + \beta^{N_2}_{(u,v)} \cdot (|B|+|C|)\right) \\
&\quad + \sum_{(u,v) \in E(A,C)} w(u,v)n \\
&= w(A)(|A|+|B|) + w(A,B)(|A|+|B|) \\
&\quad + w(C)(|B|+|C|) + w(B,C)(|B|+|C|) + w(A,C)n \\
&\quad + \frac{1}{2}w(B)(|A|+|B|+|B|+|C|) \\
&= (w(A)+w(A,B))(|A|+|B|) + (w(B,C)+w(C))(|B|+|C|) \\
&\quad + w(B)\frac{|A|+2|B|+|C|}{2} + w(A,C)n
\end{aligned}
$$

Explanation of the derivation: We classify the edges into six parts, denoted by $E(A)$, $E(A, B)$, $E(B, C)$, $E(C)$, $E(B)$, $E(A, C)$, and calculate the cost of each part separately. For $(u, v) \in E(A), E(A, B)$, the only minimal common ancestor is $N_1$, $\beta_{(u,v)}^{N_1} = 1$, $|V(N_1)| = |A| + |B|$. Similarly, for $(u, v) \in E(C), E(B, C)$, the only minimal common ancestor is $N_2$, $\beta_{(u,v)}^{N_2} = 1$, $|V(N_2)| = |B| + |C|$. For $(u, v) \in E(B)$, the minimal common ancestors are $N_1, N_2$, and $\beta_{(u,v)}^{N_1} = \beta_{(u,v)}^{N_2} = \frac{1}{2}$. For $(u, v) \in E(A, C)$, the only minimal common ancestor is $R$, $\beta_{(u,v)}^R = 1$, $|V(R)| = n$.

Observe that

$$|A| + |B|, |B| + |C|, \frac{|A| + 2|B| + |C|}{2} < n$$
$$|A| + |B| + |C| = n$$
$$w(E) = w(A) + w(B) + w(C) + w(A, B) + w(B, C) + w(A, C)$$
$$|A| + |B| = n - |C|$$
$$|B| + |C| = n - |A|$$
$$\frac{|A| + 2|B| + |C|}{2} = n - \frac{|A| + |C|}{2}$$

then we have

$$
\begin{aligned}
& cost_{primal}(A, B, C) \\
= \; & (w(A) + w(A, B))(|A| + |B|) + (w(B, C) + w(C))(|B| + |C|) \\
& + w(B)\frac{|A| + 2|B| + |C|}{2} + w(A, C)n \\
= \; & (w(A) + w(A, B))(n - |C|) + (w(B, C) + w(C))(n - |A|) \\
& + w(B)(n - \frac{|A| + |C|}{2}) + w(A, C)n \\
= \; & nw(E) - \left(w(A) + w(A, B) + \frac{w(B)}{2}\right)|C| - \left(w(C) + w(B, C) + \frac{w(B)}{2}\right)|A| \\
= \; & nw(E) - \left(w(A + B) - \frac{w(B)}{2}\right)|C| - \left(w(B + C) - \frac{w(B)}{2}\right)|A|.
\end{aligned}
$$

## B.2 PROOF OF THEOREM 3.1

We prove Theorem 3.1 with two lemmas respectively.

**Lemma B.1.** *The time complexity of Algorithm 1 is $O(\frac{n^4 \log m}{\epsilon})$.*

*Proof.* Let $w_{max}$ be the largest weight, then $w(E) \leq mw_{max}, xw(B) \leq xmw_{max}$, so $cost_{temp} = w(E) - w(A, C) + xw(B) \leq (1 + x)mw_{max}$. Observe that the edge with the largest weight is in $A$ in the initial STATE, so $cost_{temp} \geq w_{max}$. Each cycle $cost_{temp}$ increases by $1 + \frac{\epsilon}{n^2}$ times, then the maximum number of cycles is $\log_{1+\frac{\epsilon}{n^2}}((1 + x)mw_{max}/w_{max}) = O(\frac{n^2 \log m}{\epsilon})$. The time complexity is $O(\frac{n^4 \log m}{\epsilon})$. $\square$

**Lemma B.2.** *The approximate ratio of Algorithm 1 is $\frac{2}{3\sqrt{6}} - \Theta(\frac{1+\epsilon}{n})$. That is, A, B, C output by the algorithm satisfy*

$$cost_{dual} \geq (\frac{2}{3\sqrt{6}} - \Theta(\frac{1 + \epsilon}{n}))cost_{dual}(A^*, B^*, C^*),$$

*where $A^*$, $B^*$, $C^*$ are the optimal solution to 2-OC-P.*

*Proof.* Considering the conditions satisfied by $A, B, C$ at the time of termination, exchanging the nodes in $A, B$ or $B, C$ or $A, C$ at this time cannot make $cost'_{temp}(A', B', C') > (1 + \frac{\epsilon}{n^2})cost_{temp}(A, B, C)$, let $cost_{old} = cost_{temp}(A, B, C)$. In other words, if an exchange is performed again (no matter which two nodes are exchanged), $cost_{temp}$ after the exchange is denoted by

$cost_{new}$, so $cost_{new} \leq (1 + \frac{\epsilon}{n^2})cost_{old}$. Define $\Delta = cost_{new} - cost_{old}$, then we get $\Delta \leq \frac{\epsilon}{n^2}cost_{old}$. The above conclusion should be true for the exchange process of any two nodes, and the algorithm will terminate.

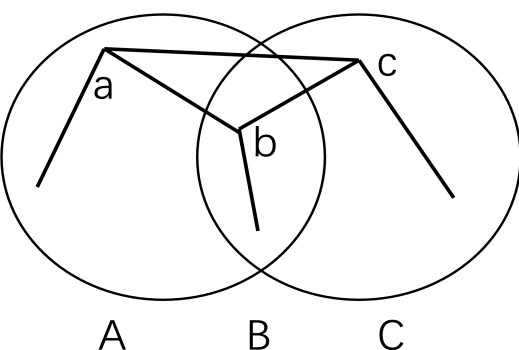

Figure 6: Schematic diagram of overlapping clustering dual problem

Before discussing the $\Delta$ value for swapping any two nodes, let's calculate some intermediate results for later use. For $a \in A, b \in B, c \in C$, denote the sets after the swap as $A'$, $B'$, $C'$, respectively. Consider the following cases.

(1) Swap $a$ and $b$: $w(A', C') - w(A, C) = w(b, C) - w(a, C)$, $w(B') - w(B) = w(a, B) - w(a, b) - w(b, B)$.

(2) Swap $b$ and $c$: $w(A', C') - w(A, C) = -w(c, A) + w(b, A)$, $w(B') - w(B) = -w(b, B) + w(c, B) - w(b, c)$.

(3) Swap $a$ and $c$: $w(A', C') - w(A, C) = -w(a, C) - w(c, A) + w(a, A) + w(c, C) + 2w(a, c)$, $w(B') - w(B) = 0$.

Note that $\Delta = -(w(A', C') - w(A, C)) + x(w(B') - w(B))$. Therefore, by substituting the above equations, we can obtain the value of $\Delta$ for each case.

Consider the memberships of the two exchanging nodes separately.

(1) For any $a \in A, b \in B$, exchanging $a, b$, we have $\Delta = cost_{new} - cost_{old} = -w(b, C) + w(a, C) + x(w(a, B) - w(a, b) - w(b, B)) \leq \frac{\epsilon}{n^2}cost_{old}$. Summing over all $a, b$, we have

$$-pnw(B, C) + (1 - 2p)nw(A, C) + x\left((1 - 2p)\, nw(A, B) - w(A, B) - 2pnw(B)\right)$$
$$\leq \quad \frac{\epsilon}{n^2}p(1 - 2p)n^2 cost_{old} \tag{4}$$

(2) For any $b \in B, c \in C$, exchanging $b, c$, by the symmetry of $A, C$, we have

$$-pnw(A, B) + (1 - 2p)nw(A, C) + x((1 - 2p)nw(B, C) - w(B, C) - 2pnw(B))$$
$$\leq \quad \frac{\epsilon}{n^2}p(1 - 2p)n^2 cost_{old} \tag{5}$$

(3) For any $a \in A, c \in C$, exchanging $a, c$, $\Delta = w(a, C) + w(c, A) - w(a, A) - w(c, C) - 2w(a, c) \leq \frac{\epsilon}{n^2}cost_{old}$. Summing over all $a, c$, we have

$$pnw(A, C) + pnw(A, C) - 2pnw(A) - 2pnw(C) - 2w(A, C) \leq \frac{\epsilon}{n^2}p^2n^2 cost_{old}$$

then

$$-w(A) - w(C) \leq -\frac{pn - 1}{pn}w(A, C) + \frac{\epsilon p^2 cost_{old}}{2pn} \tag{6}$$

Summing up Inequalities (4) and (5), we get

$$((-p + (1 - 2p)x)n - x)w(A, B) + ((-p + (1 - 2p)x)n - x)w(B, C) - 4pxnw(B)$$
$$\leq \quad (-2 + 4p)nw(A, C) + 2\epsilon p(1 - 2p)cost_{old} \tag{7}$$

Substituting $x = \frac{p}{1+2p}$ into Inequality (7), we get

$$-\frac{4p^2n+p}{1+2p}(w(A,B)+w(B,C)+w(B))+\frac{p}{1+2p}w(B)$$
$$\leq (-2+4p)nw(A,C)+2\epsilon p(1-2p)cost_{old} \tag{8}$$

Multiplying the coefficient $\frac{4p^2n+p}{1+2p}$ on Inequality (6), we get

$$-\frac{4p^2n+p}{1+2p}(w(A)+w(C)) \leq -\frac{4p^2n+p}{1+2p}\cdot\frac{pn-1}{pn}w(A,C)+\frac{4p^2n+p}{1+2p}\cdot\frac{\epsilon p^2 cost_{old}}{2pn} \tag{9}$$

Summing up Inequalities (8) and (9), we have

$$-\frac{4p^2n+p}{1+2p}w(E)+\frac{p}{1+2p}w(B) \leq \left((-2+4p)n-\frac{4p^2n+p}{1+2p}\cdot\frac{2pn-1}{pn}\right)w(A,C)$$
$$+2\epsilon p(1-2p)cost_{old}+\frac{4p^2n+p}{1+2p}\cdot\frac{\epsilon p^2 cost_{old}}{2pn} \tag{10}$$

After removing $w(B)$ on the left, we get

$$\frac{4p^2n+p}{1+2p}w(E) \geq \left((2-4p)n+\frac{4p^2n+p}{1+2p}\cdot\frac{2pn-1}{pn}\right)w(A,C)$$
$$-2\epsilon p(1-2p)cost_{old}-\frac{4p^2n+p}{1+2p}\cdot\frac{\epsilon p^2 cost_{old}}{2pn} \tag{11}$$

Focusing on $w(A,C)$, we have

$$\begin{aligned}
w(A,C) &\leq \frac{\frac{4p^2n+p}{1+2p}}{(2-4p)n+\frac{4p^2n+p}{1+2p}\cdot\frac{2pn-1}{pn}}w(E)+2\epsilon p(1-2p)cost_{old}\\
&\quad +\frac{4p^2n+p}{1+2p}\cdot\frac{\epsilon p^2 cost_{old}}{2pn}\\
&\leq \frac{4p^2n^2+pn}{2n^2-2pn-1}w(E)+\frac{2\epsilon p(1-2p)(1+x)}{(2-4p)n+\frac{4p^2n+p}{1+2p}\cdot\frac{2pn-1}{pn}}w(E)\\
&\quad +\frac{\frac{4p^2n+p}{1+2p}\cdot\frac{\epsilon p^2(1+x)}{2pn}}{(2-4p)n+\frac{4p^2n+p}{1+2p}\cdot\frac{2pn-1}{pn}}w(E)\\
&\leq \frac{4p^2n^2+pn}{2n^2-2pn-1}w(E)+\frac{2\epsilon p(1-2p)\cdot\frac{1+3p}{1+2p}}{(2-4p)n+\frac{4p^2n+p}{1+2p}\cdot\frac{2pn-1}{pn}}w(E)\\
&\quad +\frac{\frac{4p^2n+p}{1+2p}\cdot\frac{\epsilon p^2}{2pn}\cdot\frac{1+3p}{1+2p}}{(2-4p)n+\frac{4p^2n+p}{1+2p}\cdot\frac{2pn-1}{pn}}w(E)\\
&\leq \frac{4p^2n^2+pn}{2n^2-2pn-1}w(E)+\frac{2np(-6p^2+p+1)\epsilon}{2n^2-2np-1}w(E)\\
&\quad +\frac{p^2(4np+1)(3p+1)\epsilon}{2(2p+1)(2n^2-2np-1)}w(E)\\
&\leq \left(2p^2+\Theta(\frac{1}{n}))w(E)+\Theta(\frac{\epsilon}{n}\right)w(E)\\
&= \left(2p^2+\Theta(\frac{1+\epsilon}{n})\right)w(E) \tag{12}
\end{aligned}$$

So

$$w(E)-w(A,C) \geq (1-2p^2-\Theta(\frac{1+\epsilon}{n}))w(E).$$

At last,

$$
\begin{aligned}
cost_{dual}(A,B,C) &= \left(w(A+B) - \frac{w(B)}{2}\right)|C| + \left(w(B+C) - \frac{w(B)}{2}\right)|A| \\
&= (w(A+B) + w(B+C) - w(B))pn \\
&= (w(E) - w(A,C))pn \\
&\geq \left(1 - 2p^2 - \Theta(\frac{1+\epsilon}{n})\right)w(E)pn \\
&= \left(-2p^3 + p - \Theta(\frac{1+\epsilon}{n})\right)nw(E) \\
&\geq \left(-2p^3 + p - \Theta(\frac{1+\epsilon}{n})\right)cost_{dual}(A^*, B^*, C^*) \quad (13)
\end{aligned}
$$

The fact $cost_{dual} \leq nw(E)$ is used here.

Letting $p = \frac{1}{\sqrt{6}}$ and substitute it into Inequality (13), we can get

$$
cost_{dual}(A,B,C) \geq \left(\frac{2}{3\sqrt{6}} - \Theta(\frac{1+\epsilon}{n})\right)cost_{dual}(A^*, B^*, C^*) \quad (14)
$$

$\square$

Remarks: We have a rounding error in $cost_{dual}(A,B,C)$ incurred by setting $|A|, |B|, |C|$ to be integers in the above proof. But because the error is a constant multiple of $w(E)$ due to the constant errors incurred by each of $|A|, |B|, |C|$, it can be absorbed safely in $\epsilon$.

By Lemma B.1, B.2, Theorem 3.1 follows.

### B.3 PROOF OF PROPOSITION 3.2

*Proof.* We give an example satisfying that $OPT = (\frac{2}{3\sqrt{6}} - \Theta(\frac{1}{n}))nw(E)$.

Consider a complete graph with $n$ nodes, and study its optimal solution for OC-D. Consider any $A, B, C$, let $x = |A|, y = |B|, z = |C|$. Let

$$
f(x,y,z) = cost_{dual}(A,B,C) = \left(\frac{x(x-1)}{2} + xy + \frac{y(y-1)}{4}\right)z + \left(\frac{z(z-1)}{2} + yz + \frac{y(y-1)}{4}\right)x,
$$

and our goal is $\max_{x,y,z\in\mathbb{Z}_+, x+y+z=n} f(x,y,z)$.

First prove that $f(x,y,z)$ takes the maximum value when $x = y$.

Consider $(x,y,z) \rightarrow \left(\frac{x+z}{2}, y, \frac{x+z}{2}\right)$, then we have

$$
\begin{aligned}
& f\left(\frac{x+z}{2}, y, \frac{x+z}{2}\right) - f(x,y,z) \\
={}& \frac{x+z}{2}\left(\frac{x+z}{2}\left(\frac{x+z}{2}-1\right) + (x+z)y + \frac{y(y-1)}{2}\right) - \frac{x+z}{2}\frac{y(y-1)}{2} \\
& -2xyz - \frac{xz(x+z-2)}{2} \\
={}& \frac{x+z-2}{2}\left(\frac{(x+z)^2 - 4xz}{4}\right) + \frac{(x+z)^2 - 4xz}{2}y \\
\geq{}& 0
\end{aligned}
$$

indicating that when $y$ is given, $f$ gets maximum when $x = z$.

So the problem is transformed into $\max_{x,y\in\mathbb{Z}_+, 2x+y=n} g(x,y) = \left((x-1)x + 2xy + \frac{y(y-1)}{2}\right)x$, and is further transformed into

$$
\max_{0<x<\frac{n}{2}} h(x) = \left((x-1)x + 2x(n-2x) + \frac{(n-2x)(n-2x-1)}{2}\right)x
$$

Then we have

$$h(x) = \left((x-1)x + (n-2x)\frac{n-1+2x}{2}\right)x$$

$$= \left((x-1)x + \frac{n^2 - 4x^2 - (n-2x)}{2}\right)x$$

$$h'(x) = -3x^2 + \frac{n^2 - n}{2}$$

So, $h(x)$ gets the maximum value at $x = \sqrt{\frac{n^2 - n}{6}}$. For the optimal solution value, then we have

$$cost^*_{dual} = -\frac{n^2 - n}{6}\sqrt{\frac{n^2 - n}{6}} + \frac{n^2 - n}{2}\sqrt{\frac{n^2 - n}{6}}$$

$$= \frac{n^2 - n}{3}\sqrt{\frac{n^2 - n}{6}}$$

Finally, notice that $nw(E) = n \cdot \frac{n(n-1)}{2} = \frac{n^3 - n^2}{2}$. Then we have

$$\frac{cost^*_{dual}}{nw(E)} = \frac{\frac{n^2 - n}{3}\sqrt{\frac{n^2 - n}{6}}}{\frac{n^3 - n^2}{2}}$$

$$= \frac{2}{3}\sqrt{\frac{n^2 - n}{6n^2}}$$

$$= \frac{2}{3\sqrt{6}}\frac{\sqrt{n^2 - n} + n - n}{n}$$

$$= \frac{2}{3\sqrt{6}} - \frac{2}{3\sqrt{6}}\frac{n - \sqrt{n^2 - n}}{n}$$

$$= \frac{2}{3\sqrt{6}} - \frac{2}{3\sqrt{6}}\frac{n}{n(n + \sqrt{n^2 + n})}$$

$$= \frac{2}{3\sqrt{6}} - \frac{2}{3\sqrt{6}}\frac{1}{n + \sqrt{n^2 + n}}$$

$$= \frac{2}{3\sqrt{6}} - \epsilon$$

where $\epsilon = \Theta\left(\frac{1}{n}\right)$.

It can be observed that, for a complete graph, its relationship with $nw(E)$ is exactly the approximate ratio of Algorithm 1. □

### B.4 APPROXIMATION GUARANTEE FOR 2-OC-P

In the section, we show that Algorithm 1 is actually a good approximation algorithm for 2-OC-P. We simply treats the output of Algorithm 1 as the result for 2-OC-P, we have the following approximation guarantee for 2-OC-P.

**Theorem B.3.** *The approximation factor of Algorithm 1 for 2-OC-P is* $(1-a)(1 + d_{max}/d_{avg})$, *where $d_{max}$ is the maximum degree of all nodes, $d_{avg}$ is the average degree, and $a = \frac{2}{3\sqrt{6}} - \Theta(\frac{1+\epsilon}{n})$.*

By this theorem, we have two corollaries for regular and bounded-degree graphs, respectively.

**Corollary B.4.** *If the graph $G$ is $d$-regular, then the approximation factor of Algorithm 1 for 2-OC-P on $G$ is $(1-a)(1+d)$.*

**Corollary B.5.** *If the degree of each node in graph $G$ is upper bounded by a certain constant $d$, then the approximation factor of Algorithm 1 for 2-OC-P on $G$ is $2(1-a)d$.*

Then we prove Theorem B.3.

*Proof.* Let $cost^*_{dual}$ and $cost^*_{primal}$ be the optimal objective values, $cost_{dual}$ and $cost_{orimal}$ be the values that Algorithm 1 outputs, for 2-OC-D and 2-OC-P, respectively. Let $a$ and $b$ be the approximate ratios of Algorithm 1 for the dual and the primal problems, respectively. We have the following relationship.

**Lemma B.6.** $b \leq (1-a)(1 + cost^*_{dual}/cost^*_{primal})$.

*Proof.* Note the following relationships hold.

$$cost^*_{primal} + cost^*_{dual} = cost_{primal} + cost_{dual} = n \cdot w(E)$$

and

$$cost_{dual} \geq anw(E) \geq a \cdot cost^*_{dual}$$

Therefore, we have

$$
\begin{aligned}
cost_{primal} &= nw(E) - cost_{dual} \\
&\leq nw(E) - anw(E) \\
&= (1-a)(cost^*_{primal} + cost^*_{dual})
\end{aligned}
$$

Since $cost_{primal} = b \cdot cost^*_{primal}$, Lemma B.6 follows. $\qquad\square$

Since we already have $a = \frac{2}{3\sqrt{6}} - \Theta(\frac{1}{n})$, if we can give an upper bound on $cost^*_{dual}/cost^*_{primal}$, then we also have an upper bound on $b$. The following lemma provides this upper bound.

**Lemma B.7.** *For* 2-*OC-D and* 2-*OC-P,* $cost^*_{dual}/cost^*_{primal} < \rho_{max}/\rho_{avg} \leq d_{max}/d_{avg}$, *where* $\rho_{avg} = w(E)/|V|$ *is the average density,* $\rho_{max}$ *is the maximum density of all induced subgraphs,* $d_{max}$ *is the maximum degree of all nodes, and* $d_{avg}$ *is the average degree of all nodes.*

*Proof.* Let $A^*$, $B^*$, $C^*$ be the optimal solution (no matter whether it is primal or dual because the two problems are equivalent), let $cost^*_{primal}$ denote $cost_{primal}(A^*, B^*, C^*)$ and $cost^*_{dual}$ denote $cost_{dual}(A^*, B^*, C^*)$ for short. Let $A, B, C$ be the output of Algorithm 1, and use it as the output for 2-OC-P. Let $cost_{primal}$ denote $cost_{primal}(A, B, C)$ and $cost_{dual}$ denote $cost_{dual}(A, B, C)$.

$$
\begin{aligned}
cost^*_{dual} &= |C^*| \left( w(A^*) + w(A^*, B^*) + \frac{1}{2}w(B^*) \right) \\
&\quad + |A^*| \left( w(C^*) + w(B^*, C^*) + \frac{1}{2}w(B^*) \right) \\
cost^*_{primal} &= (|A^*| + |B^*|) \left( w(A^*) + w(A^*, B^*) + \frac{1}{2}w(B^*) \right) \\
&\quad + (|B^*| + |C^*|) \left( w(C^*) + w(B^*, C^*) + \frac{1}{2}w(B^*) \right) + nw(A^*, C^*).
\end{aligned}
$$

Consider two cases of $\frac{cost^*_{dual}}{cost^*_{primal}}$.

(1) If $\frac{cost^*_{dual}}{cost^*_{primal}} \leq 1$, then $b = 2(1-a)$.

(2) If $\frac{cost^*_{dual}}{cost^*_{primal}} > 1$, namely $cost^*_{dual} > cost^*_{primal}$, substituting into the specific form of the objective function, we have

$$
\begin{aligned}
&|C^*| \left( w(A^*) + w(A^*, B^*) + \frac{1}{2}w(B^*) \right) + |A^*| \left( w(C^*) + w(B^*, C^*) + \frac{1}{2}w(B^*) \right) \\
> \ &(|A^*| + |B^*|) \left( w(A^*) + w(A^*, B^*) + \frac{1}{2}w(B^*) \right) \\
&+ (|B^*| + |C^*|) \left( w(C^*) + w(B^*, C^*) + \frac{1}{2}w(B^*) \right) + nw(A^*, C^*).
\end{aligned}
$$

Then we get

$$(|C^*| - |A^*| - |B^*|)\left(w(A^*) + w(A^*, B^*) + \frac{1}{2}w(B^*)\right)$$

$$+ (|A^*| - |B^*| - |C^*|)\left(w(C^*) + w(B^*, C^*) + \frac{1}{2}w(B^*)\right)$$

$$> nw(A^*, C^*).$$

This holds if and only if

$$(|C^*| - |A^*|)(w(A^*) + w(A^*, B^*) - w(C^*) - w(B^*, C^*))$$
$$> nw(A^*, C^*) + |B^*|(w(A^*) + w(A^*, B^*) + w(B^*) + w(C^*) + w(B^*, C^*)),$$

which implies that

$$(|C^*| - |A^*|)(w(A^*) + w(A^*, B^*) - w(C^*) - w(B^*, C^*)) > 0.$$

Without loss of generality, we assume that $|C^*| > |A^*|$, $w(A^*) + w(A^*, B^*) > w(C^*) + w(B^*, C^*)$. We consider two types of scaling for $\frac{cost^*_{dual}}{cost^*_{primal}}$.

Scale 1: replace $|A^*|$ with $|C^*|$ for the numerator, remove $nw(A^*, C^*)$ for the denominator, replace $|C^*|$ with $|A^*|$, and then we get

$$\frac{cost^*_{dual}}{cost^*_{primal}}$$

$$< \frac{|C^*|(w(A^*)+w(A^*,B^*)+\frac{1}{2}w(B^*))+|C^*|(w(C^*)+w(B^*,C^*)+\frac{1}{2}w(B^*))}{(|A^*|+|B^*|)(w(A^*)+w(A^*,B^*)+\frac{1}{2}w(B^*))+(|B^*|+|A^*|)(w(C^*)+w(B^*,C^*)+\frac{1}{2}w(B^*))}$$

$$= \frac{|C^*|(w(A^*)+w(A^*,B^*)+w(B^*)+w(C^*)+w(B^*,C^*))}{(|A^*|+|B^*|)(w(A^*)+w(A^*,B^*)+w(B^*)+w(C^*)+w(B^*,C^*))}$$

$$= \frac{|C^*|}{|B^*|+|A^*|}.$$

Scale 2: replace the numerator $w(C^*) + w(B^*, C^*)$ with $w(A^*) + w(A^*, B^*)$, remove the denominator Contents of $|B^*|$, and then we have

$$\frac{cost^*_{dual}}{cost^*_{primal}}$$

$$< \frac{(|A^*|+|C^*|)(w(A^*)+w(A^*,B^*)+\frac{1}{2}w(B^*))}{|A^*|(w(C^*)+w(B^*,C^*)+\frac{1}{2}w(B^*))+|C^*|(w(C^*)+w(B^*,C^*)+\frac{1}{2}w(B^*))+(|A^*|+|C^*|)w(A^*,C^*)}$$

$$< \frac{(|A^*|+|C^*|)(w(A^*)+w(A^*,B^*)+\frac{1}{2}w(B^*))}{(|A^*|+|C^*|)(w(C^*)+w(B^*,C^*)+\frac{1}{2}w(B^*)+w(A^*,C^*))}$$

$$= \frac{w(A^*)+w(A^*,B^*)+\frac{1}{2}w(B^*)}{w(C^*)+w(B^*,C^*)+\frac{1}{2}w(B^*)+w(A^*,C^*)}.$$

Therefore,

$$\frac{cost^*_{dual}}{cost^*_{primal}} < \max\left\{1, \min\left\{\frac{|C^*|}{|A^*|+|B^*|}, \frac{w(A^*) + w(A^*, B^*) + \frac{1}{2}w(B^*)}{w(C^*) + w(B^*, C^*) + \frac{1}{2}w(B^*) + w(A^*, C^*)}\right\}\right\}$$

Let

$$x = \frac{|C^*|}{|A^*| + |B^*|}$$

$$y = \frac{w(A^*) + w(A^*, B^*) + \frac{1}{2}w(B^*)}{w(C^*) + w(B^*, C^*) + \frac{1}{2}w(B^*) + w(A^*, C^*)}$$

then we have

$$|A^*| + |B^*| = \frac{1}{1+x}|V|$$

and

$$w(A^*) + w(A^*, B^*) + \frac{1}{2}w(B^*) = \frac{y}{1+y}w(E)$$

Recall that $|C^*| > |A^*|$, $w(A^*) + w(A^*, B^*) > w(C^*) + w(B^*, C^*)$, and observe that the density of the induced subgraph $G[A^* + B^*]$ should be large. Set $\rho_{max} = \max_{U \subseteq V}\left\{\frac{w(E(G[U]))}{|U|}\right\}$ to be the

maximum density of the induced subgraph on $G$, $E(G[U])$ to be the edge set of $G[U]$, $\rho_{avg} = \frac{w(E)}{|V|}$ to be the average density of $G$, and then

$$\frac{w(A^*) + w(A^*, B^*) + \frac{1}{2}w(B^*)}{|A^*| + |B^*|}$$

$$= \frac{y(1+x)}{1+y} \cdot \frac{w(E)}{|V|}$$

$$\leq \frac{w_{A+B}}{|A^*| + |B^*|}$$

$$\leq \rho_{max}$$

We have

$$\frac{y(1+x)}{1+y} \leq \frac{\rho_{max}}{\rho_{avg}}$$

Now according to the value of $min(x, y)$, we consider the following two cases.

(1)$x \leq y$ :, we have

$$x = \frac{(1+y)x}{1+y} = \frac{x+xy}{1+y} \leq \frac{y+xy}{1+y} = \frac{y(1+x)}{1+y} \leq \frac{\rho_{max}}{\rho_{avg}}$$

(2)$x > y$, we have

$$y = \frac{y(1+x)}{1+x} < \frac{y(1+x)}{1+y} \leq \frac{\rho_{max}}{\rho_{avg}}$$

Therefore,

$$\min\left\{\frac{|A^*|}{|B^*|}, \frac{w(B^*)}{w(A^*) + w(A^*, B^*)}\right\} < \frac{\rho_{max}}{\rho_{avg}}$$

Then for $\frac{cost^*_{dual}}{cost^*_{primal}}$, we have

$$\frac{cost^*_{dual}}{cost^*_{primal}} < \max(1, \frac{\rho_{max}}{\rho_{avg}}) = \frac{\rho_{max}}{\rho_{avg}}$$

Calculating $\rho_{max}$ is difficult. However, it is observed that the average degree $d \leq d_{max}$ on $G[U]$, and $w(E(G[U])) = \frac{d|U|}{2}$. So, we have an upper bound on $\rho_{max}$, that is

$$\rho_{max} = \frac{w(E(G[U]))}{|U|} \leq \frac{d \cdot |U|}{2|U|} \leq \frac{d_{max}}{2}$$

On the other hand,

$$\rho_{avg} = \frac{w(E)}{|V|} = \frac{d_{avg} \cdot |V|}{2|V|} = \frac{d_{avg}}{2}$$

This implies that

$$\frac{\rho_{max}}{\rho_{avg}} \leq \frac{d_{max}}{d_{avg}}$$

Therefore, $\frac{cost^*_{dual}}{cost^*_{primal}} < \frac{\rho_{max}}{\rho_{avg}} \leq \frac{d_{max}}{d_{avg}}$, approximate ratio $b = (1-a)\left(1 + \frac{d_{max}}{d_{avg}}\right)$. □

Combining Lemmas B.6 and B.7, Theorem B.3 follows.

□

### B.5 Derivation process for approximation ratio of 2-OC-P

Note the following relationships

$$cost^*_{primal} + cost^*_{dual} = cost_{primal} + cost_{dual} = n \cdot w(E)$$

and

$$cost_{dual} \geq anw(E) \geq a \cdot cost^*_{dual}$$

Therefore, we have

$$
\begin{aligned}
cost_{primal} &= nw(E) - cost_{dual} \\
&\leq nw(E) - anw(E) \\
&= (1-a)(cost^*_{primal} + cost^*_{dual}) \\
&= b \cdot cost^*_{primal}
\end{aligned}
$$

### B.6 Proof of Theorem 3.3

*Proof.* Note that Algorithm 1 achieves a dual cost at least $\left(\frac{2}{3\sqrt{6}} - \Theta(\frac{1+\epsilon}{n})\right) \cdot nw(E)$ for 2-OC-D, in which $nw(E)$ is an upper bound for the cost of the dual HOC problem with any constraints. So, Theorem 3.3 follows if the final dual cost is no less than the one after the first round of invoking Algorithm 1.

Let $N_1$ and $N_2$ be the two overlapping clusters that Algorithm 1 yields in the first round of the repeat loop, and $e \in E$ be an edge that treats $N_1$ or $N_2$ as a common ancestor. Then the root $r$ is not a minimal common ancestor of $e$, since otherwise, $r$ forms an chain with $N_1$ or $N_2$. In the next iterations, $r$ will not be included in the minimal common ancestor set of $e$ during both splitting and merging process. Since $N_1$ and $N_2$ have the same size, no matter how the belonging factors of $e$ change, the final primal cost that $e$ contributes will not exceed that after the first round of invoking Algorithm 1. Theorem 3.3 follows immediately. □

### B.7 Pseudocode of the speed-up algorithm

We present the pseudocode of the speed-up algorithm for 2-OC in Algorithm 3.

---

**Algorithm 3:** Speed-up algorithm for 2-OC

---

**Input:** an undirected graph $G = (V, E, w)$, move batch ratio $\gamma$
**Output:** node sets $A$, $B$ and $C$ for 2-OC
1 $X, Y \leftarrow RatioCut(G)$;
2 $A \leftarrow X, B \leftarrow \emptyset, C \leftarrow Y$;
3 **repeat**
4      Calculate the delta of $cost_{dual}$ when each node moves to the other two sets, and select the one with the larger increment as the potential action at that node;
5      Let $S$ be the node set that brings $cost_{dual}$ increment;
6      $t \leftarrow \min\{|S|, \gamma|V|\}$;
7      Move the top-$t$ nodes with the largest increment;
8 **until** $S$ *is empty*;
9 return $A$, $B$, $C$.

---

Replacing Algorithm 1 with Algorithm 3 in Algorithm 2, we get the speed-up version of $k$-HOC algorithm.

## C Supplement to Experiments

In this section, we provide more information about our experiments.

### C.1 OSBM AND OUR SETTINGS

OSBM is specified by a $k \times k$ symmetric matrix $Z$, where each element is a natural number, and a pair of real numbers $p_1$, $p_2$ ($0 \leq p_1 \leq p_2 \leq 1$). $Z_{ij}$ ($i \neq j$) represents the number of overlapping nodes between the $i$-th and the $j$-th clusters, and $Z_{ii}$ represents the number of nodes in the $i$-th cluster that do not participate in overlap. Denote by $C_1, ...C_k$ the planted overlapping clusters. $p_1$ represents the inter-link probability between each pair of clusters, while $p_2$ represents the intra-link probability within each cluster. For two nodes in the overlapping parts, we have two independent samples, and the edge is present if any of them generate an edge. In other words, the probability of edge presence between any two nodes in the overlapping parts is $1 - (1 - p_2)^2$.

In our experiments, we assume that each node belongs to at most two clusters. Thus, the total number of nodes is the sum of entries in the upper triangle of $Z$. For simplicity of implementation, all clusters are of the same size and have the same size of overlaps between clusters. For a 2-level hierarchical structure, we choose three probability values $0 \leq p_1 \leq p_2 \leq p_3 \leq 1$, in which $p_1$ is the inter-link probability between clusters on the first level, $p_2$ is the inter-link probability between clusters on the second level, and $p_3$ is the intra-link probability within each cluster. So now, the probability of edge presence between any two nodes in the overlapping parts is $1 - (1 - p_3)^2$.

### C.2 DEFINITION OF NMI FOR OC

NMI for OC is a natural generalization from NMI for non-overlapping partition. For two different groups of overlapping clusters $X = \{x_1, x_2, ...\}, Y = \{y_1, y_2, ...\}$ on the same graph $G = (V, E)$ , $x_i, y_i$ are all clusters. We define NMI of $X, Y$ as follows.

$$p(x_i) = \frac{|x_i|}{|V|}$$

$$p(x_i, y_j) = \frac{|x_i \cap y_j|}{|V|}$$

$$H(X) = - \sum_{x_i \in X} p(x_i) \log p(x_i)$$

$$H(X, Y) = - \sum_{x_i \in X, y_j \in Y} p(x_i, y_j) \log p(x_i, y_j)$$

$$I(X : Y) = H(X) + H(Y) - H(X, Y)$$

$$NMI(X, Y) = \frac{I(X : Y)}{\max(H(X), H(Y))}$$

### C.3 EVALUATION ON THE MNIST DATASET

To show intuitively that our algorithm is able to find out the blurred overlapping area of datasets, we run our 2-OC algorithm on the MNIST dataset LeCun et al. (1998), which is a benchmark of handwritten digits containing ten classes of images labeled by $0 \sim 9$, respectively. We select two pairs of labels that are easily confused by hand writing, i.e., 1 vs. 7, 3 vs. 8, and construct a $k$-nearest neighbor graph for each of them. Each node of the graph represents an image of handwritten digit, and the similarity is measured by applying the Gaussian kernel function to the Euclidean distance of pixel vectors. We remark that not all embeddings (e.g., word embeddings) that are generated by modern-day AI models are suitable for clustering. We just find that pixel vector in MNIST is somewhat a good use-case to showcase our results of overlapping, ambiguous samples.

The parameters, NMI, size of the overlapping part, the costs of ground truth (GT) and 2-OC output are summarized in Table 5. NMI is calculated with the non-overlapping ground truth of data points, although our algorithm gives overlapping results. However, the NMI for the labels 1 vs. 7 is above 0.9, and only 5 digits, which can be viewed as ambiguous ones, are allocated in the overlapping part. We demonstrate all of them in Figure 7(a). For the labels 3 vs. 8, there are 180 ambiguities. We demonstrate five of them in Figure 7(b). A significant factor that impacts the accuracy of our algorithm is that we simply use the pixel vectors of digits which is a very rough representation of images.

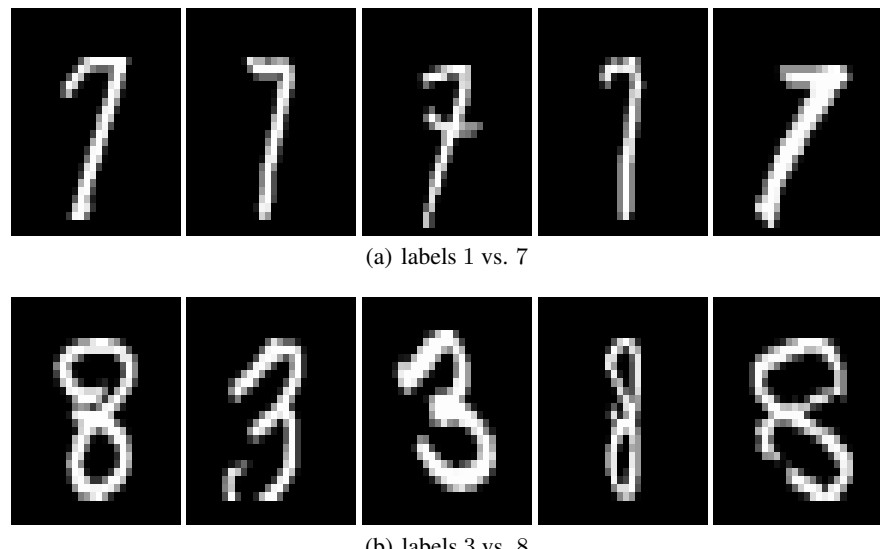

(a) labels 1 vs. 7

(b) labels 3 vs. 8

Figure 7: Demonstrations of the ambiguous samples our 2-OC algorithm yields.

Table 5: Parameters and results on the MNIST dataset

| label | size | $k$ | NMI | overlapping size | GT cost | 2-OC cost |
|---|---|---|---|---|---|---|
| $1\ vs.\ 7$ | $7877 + 7293$ | 100 | 0.915 | 5 | $7.37 \times 10^9$ | $7.33 \times 10^9$ |
| $3\ vs.\ 8$ | $7141 + 6825$ | 100 | 0.714 | 180 | $6.47 \times 10^9$ | $6.63 \times 10^9$ |

### C.4 VISUALIZATION ON FOUR OVERLAPPING CLUSTERS

We visualize in Figure 8 a 4-HOC results of Algorithm 2 on a small graph that is generated from OSBM and contains 100 nodes and 1264 edges. It has four embedded overlapping clusters of size 30, each of which contains 20 nodes that entirely belong to the cluster. There are 6 overlapping regions, each of which corresponds to a pair of overlapping clusters out of the 4 clusters, and each region contains 2 nodes. We label them from 81 to 100. We demonstrate the ground-truth membership of all nodes in Tables 6 and 7. The edge presence probabilities are $p_1 = 0.05$, $p_2 = 0.1$ and $p_3 = 0.5$.

Table 6: Membership of level-2 nodes in each of the four clusters. Each diagonal entry numbers the nodes that belong exclusively to the corresponding cluster. The entry $(i, j)$ $(i \neq j)$ denotes the overlapping region between clusters $i$ and $j$. In the visualization, the corresponding colors of clusters 1, 2, 3, and 4 are red, green, yellow, and blue, respectively, while the overlapping nodes are the mixed colors of their clusters.

| cluster label | 1 | 2 | 3 | 4 |
|---|---|---|---|---|
| 1 | 1-20 | 85,91,97 | 86,92,98 | 81,87,93,99 |
| 2 | 85,91,97 | 21-40 | 82,88,94,100 | 83,89,95 |
| 3 | 86,92,98 | 82,88,94,100 | 41-60 | 84,90,96 |
| 4 | 81,87,93,99 | 83,89,95 | 84,90,96 | 61-80 |

Table 7: Membership of level-1 nodes in each of the two clusters. Clusters 1 and 2 form one cluster, denoted by $(1, 2)$, on level 1, while clusters 3 and 4 form the other one, denoted by $(3, 4)$.

| cluster label | (1,2) | (3,4) |
|---|---|---|
| (1,2) | 1-40,85,91,97, | 81-83,86-89,92-95,98-100 |
| (3,4) | 81-83,86-89,92-95,98-100 | 41-80,84,90,96 |

Our algorithm bipartitions the node set at the first level into two overlapping clusters, one consists of red and green (1 and 2), the other yellow and blue(3 and 4). It achieve NMI = 0.881 on this level.

Table 8: List of misclassified nodes given by our algorithm on level 2.

| node number | cluster label in ground truth | cluster label by the $k$-HOC algorithm |
|---|---|---|
| 81 | 1,4 | 4 |
| 88 | 2,3 | 3 |
| 98 | 1,3 | 3 |
| 99 | 1,4 | 4 |

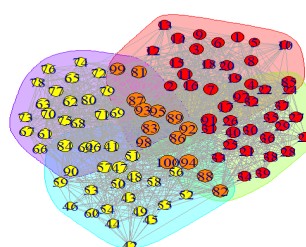 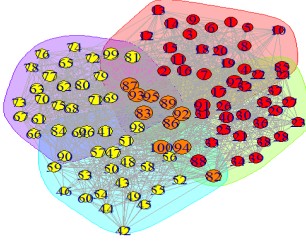

(a) Ground truth of the 4 clusters (Level 1). (b) The result of our algorithm (Level 1).

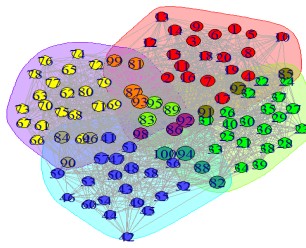 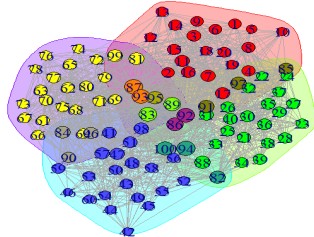

(c) Ground truth of the 4 clusters (Level 2). (d) The result of our algorithm (Level 2).

Figure 8: Visualization of a 4-HOC clustering.

Table 9: List of misclassified nodes given by our algorithm on level 1.

| node number | cluster label in ground truth | cluster label by the $k$-HOC algorithm |
|---|---|---|
| 81 | (1,2),(3,4) | (3,4) |
| 88 | (1,2),(3,4) | (1,2) |
| 98 | (1,2),(3,4) | (3,4) |
| 99 | (1,2),(3,4) | (3,4) |

At the second level, it achieves NMI $= 0.914$ for the 4 ground-truth overlapping clusters. In Figure 8, we visualize this result. Our algorithm successfully captures the overall outlines of the clusters, except membership errors on only four nodes, whose labels are 81, 88, 98 and 99. We list them in Tables 9 and 8. These nodes are misclassified into non-overlapping region.

