# OpenReview forum: "Hierarchical overlapping clustering: cost function, algorithm and scalability"
_ICLR.cc/2025/Conference — Submitted to ICLR 2025_

### Official Review · Reviewer_XQNF · 2024-10-30

**Soundness:** 3
**Presentation:** 2
**Contribution:** 4
**Rating:** 8
**Confidence:** 4

**Summary:**

The paper introduces the first cost function for hierarchical clustering with overlap.
The cost function is a natural extension of DasGupta for the case of overlap.
The authors provide a polynomial time algorithm that achieves as constant factor approximation (under some assumptions on the overlap) for the dual. The authors complement the theory results with some experiments.

**Strengths:**

I think this is a great model (combining hierarchical clustering and overlap). To make it clear, the model is the main contribution for me.

The cost function is very natural

The algorithms seem natural.

There are rigorous theory results.

**Weaknesses:**

The presentation can be improved. A running example for all the definitions would be extremely useful. Also, give intuitions for all the definitions.

The definition of 2-OC needs to be better motivated and made much clearer.

Talk more about the primal and dual version.

**Questions:**

Are you willing to add an example (can be in the supplementary material) for all definitions?

---

> ### Author Response · Authors · 2024-11-18
> **Rebuttal**
>
> Thank the reviewer for the valuable comments. We address the concerns as follows.
>
> W1) We totally agree that a running example for all definitions would be extremely useful. For a newly defined problem, there are always too many things to introduce. When we prepared this manuscript, we hope to include as many as possible in the constrained space. So, we move some intuitive examples to the appendix.
>
> W2) OK, we use 2-OC algorithm as a subroutine for the $k$-HOC problem, which is our main purpose in this paper. However, we believe that our 2-OC cost has its own interests since it is a counterpart of Orecchia et al. (2022)’s work that is the baseline competitor in the scalability experiment.
>
> W3) The primal and dual versions of HOC problem are similar to those in the scenario of HC, since for both problems, the sum of the primal and dual cost is fixed. There have been plenty of studies on both primal and dual of HC cost, but not on HOC yet. So, we believe that there are also many problems that worth study along these two lines for HOC.
>
> Q1) Yes, of course. We would like to add another section in the supplementary material (and also adjust the contents in the main body) to illustrate all the definitions.

---

> > ### Comment · Reviewer_XQNF · 2024-11-19
> >
> > Thank you for the clarifications, much appreciated.

---

### Official Review · Reviewer_c9fX · 2024-10-30

**Soundness:** 3
**Presentation:** 2
**Contribution:** 2
**Rating:** 5
**Confidence:** 3

**Summary:**

This paper studies hierarchical overlapping clustering (HOC), in which vertices are assigned to a hierarchical structure of overlapping clusters. In comparison with non-overlapping HC, we construct a DAG rather than an HC tree.

The paper introduces an objective function for this problem, generalising Dasgupta's cost function for HC, and gives a constant-factor approximation algorithm for the dual objective. Finally, the paper includes some experimental evaluation and compares the algorithm with non-hierarchical overlapping clustering algorithms, finding that the new algorithm has a faster running time.

**Strengths:**

The paper introduces a new problem which has not been previously studied: hierarchical overlapping clustering, thus the whole paper is quite original. The paper is broadly readable and clear. The given objective function extends Dasgupta's cost in a sensible way to the case with overlapping clusters. The given algorithm is quite practical, performing well on synthetic data, and with a fast running time on large real-world datasets.

**Weaknesses:**

1. The motivation for HOC is not very clear. It would be helpful to discuss a little more about cases and applications where HOC would be useful where OC or HC does not already suffice.
2. I found the technical description of the HOC graph and the objective function quite difficult to follow. In particular, the definition of parent / child nodes seems non-standard as the edges are oriented from the children towards the parents (or should in-degree and out-degree be swapped in definition 2.1?)
3. The main theoretical result is the approximation algorithm for the dual problem (Theorem 3.7). However, I am a little unsure about the significance of this algorithm. In particular, it is shown that an upper bound of the objective is $n w(E)$ and the proof of Theorem 3.7 shows only that the result of the algorithm has objective at least $\alpha n w(E)$, where $\alpha$ is the claimed approximation guarantee. Thus the guarantee on the output of the algorithm does not depend on the optimal objective value of the input. The result would be much stronger if the approximation was proved directly with respect to the optimal objective of the input.
4. The paper could be further improved with more discussion on the computational complexity of optimising the proposed objective function: it is known that Dasgupta's objective is NP-hard to optimise. Is the same true for the new objective?

**Questions:**

1. Is there hope to prove that the approximation algorithm performs better on instances with a higher objective?
2. Is the new objective function NP-hard to optimise?

---

> ### Author Response · Authors · 2024-11-18
> **Rebuttal**
>
> Thank the reviewer for the valuable comments. We address the concerns one by one.
>
> W1) Due to the space constraint, we only mentioned on Lines 33-36 two possible applications in social and cooperation networks. In our opinion, although we haven’t had a complete example to demonstrate the superiority of HOC structure, since HOC better reflects the real organization of datasets than HC and OC, the study on HOC is significant.
>
> W2) The technical description of the HOC graph and the objective function is generalized from HC trees. Since there are several new concepts, such as belongs factors, minimal common ancestor set, anti-chain, etc., it is really a bit difficult to follow. We will improve the description clearer in further version. We define the directed edges to point upwards like a rooted tree, because we often need to find the minimal common ancestor set of two leaves, which may be convenient if we define it like this.
>
> W3) Yes, a stronger upper bound than $nw(E)$ for the optimal dual cost would be much helpful to improve the approximation factor. If we can show that $OPT \leq \beta nw(E)$, then combining with the factor $\alpha$ we already have, we can get a stronger guarantee $\alpha / \beta$. However, such a non-trivial $\beta$ seems not easy to have, and we think it is a problem of independent interest.
>
> W4) The hardness of $k$-HOC problem is still open. We only know that it is easy when $k$ is as large as the number of edges. For smaller $k$, we don’t know, even if $k=2$.
>
> Q1) Do you mean to have a better guarantee on the graphs that have high objective, or say, on well overlapping clustered graphs? There is some work for HC problem on well clustered graphs, and having similar results for HOC seems interesting also. We have no idea about this problem.
>
> Q2) Hardness of $k$-HOC is open yet.

---

> > ### Comment · Reviewer_c9fX · 2024-11-26
> > **Response to rebuttal**
> >
> > I'm grateful to the authors for their response. In my view, the primary weakness of the paper is W3 in my original review, which is that the approximation guarantee in Theorem 3.7 is not particularly strong. For this reason, I will keep my original score.

---

> > > ### Author Response · Authors · 2024-11-26
> > > **Response to comments**
> > >
> > > Thank the reviewer. I will target to improve the approximation factor in our future work.

---

### Official Review · Reviewer_ZUCp · 2024-11-01

**Soundness:** 3
**Presentation:** 3
**Contribution:** 3
**Rating:** 6
**Confidence:** 4

**Summary:**

This work introduces and studies the hierarchical overlapping clustering (HOC) problem. In the clustering literature, many works have focussed on either (i) overlapping or (ii) hierarchical clustering; this work's aim is to reconcile both topics.

As a first contribution -- inspired by the well-known Dasgupta cost function -- this paper introduces a _cost function_ for overlapping hierarchical clustering. The proposed cost function has serveral desirable properties such as compatibility, additivity, and binary optimality.

As a second contribution, the paper further proposes approximation algorithms for the dual and primal variants of the new cost function. These approximation algorithms hold for a restricted variant of the HOC problem, namely the $k$-HOC problem and the $2$-HOC problem.

As a final contribution, the algorithm is tested on synthetic and real-world dataset. To speed up the algorithm, the paper uses local search heuristics.

**Strengths:**

**S1)** The paper studies a new important problem, that is worth to study. The problem formulation itself is quite intuitive (although rather difficult to grasp from the current writeup, see W1), and a natural way to combine the hierarchical and overlapping clustering problems. This work could potentially lay the groundwork for future algorithms to be developed on. Throughout the main body and appendix the paper also states and proves intuitive properties of the cost function, which is a nice contribution.

**S2)** The paper also introduces 2 approximation algorithms for the new objective function, one for the $2$-OC problem, and another $k$-HOC problem. These algorithms could provide a good starting point for future work on HOC.

**S3)** Finally, the theory for the cost function and the proofs for the approximation guarantees seem sound to me - and non-trivial for the most part.

**Weaknesses:**

**W1)** The current writeup explaining the cost function is difficult to understand. Although the proposed objective function is intuitive, it is more involved than Dasgupta's cost, and thus requires clear explanation. With the way it's currently written in the first 10 pages, I found it difficult to understand. Lines 162-171 are densely written with non-intuitative explanations. To be more specific, the sentence "(...) they are possible unrelated in meaning and clustered by different mechanism (...)" is confusing. Later, when defining the node-to-node belonging factor, on line 191 it states: "To better understand the belonging factor, it is easy to verify that the above definition is equivalent to the following plain one". The following plain definition after is not plain at all, and contains ambiguous phrasing such as "keeps the same for the other two cases".

In the remark starting on page 240, the paper introduces the notion of "width" for HOC, however it is not clear to me exactly what the width is (See Q1). The explanation here is also difficult for me to understand; "intuitively, it can be considered as a set of key clusters that are close to the leaves on the HOC graph" is an ambiguous explanation.

I think the paper would greatly benefit from having a figure in the main body. In the appendix, Figures 3 & 4 provide a much clearer picture of HOC (and the belonging factors), and also very nicely illustrate in what settings HOC makes more sense than HC. The reason I emphasise this point is because if one of the key contributions of this work is to introduce a new cost function that will be used widely, it should be easy to understand for the community


**W2)** The approximation guarantees are given on fairly restricted settings; for the primal and dual variants of $2$-OC and $k$-HOC. the $2$-OC problem studies the overlapping bipartition problem, and contains no hierarchical structure. The $k$-HOC problem does have a hierarchical component, but restricts itself to at most $k$ clusters/nodes. The approximation guarantee on the latter is only given on the dual variant of the problem (which in general is a bit easier to prove, like the dual of Dasgupta's cost function (Mosely & Wang, 2017)). It seems that the approximation guarantee follows rather directly from the approximation guarantee of the dual of $2$-OC.

**W3)** This is a minor weakness: the experiment section is slightly dis-coupled from the main body of the work

**a)** The proposed algorithms are rather slow ($O(n^4)$ factor in the running time) which prohibits the algorithm from being applied at a large scale. This requires the paper to introduce local search heuristics to make the algorithms implementable. As such, Algorithm 2 is not 'formally' compared.

**b)** $k$-HOC experiments are only performed on synthetic data, and it is not clear to me how some metrics (such as NMI) are computed for overlapping settings for both methods compared (See Q4).

**c)** No qualitatative results on real-world datasets; the real-world datasets are only used for scalability experiments. There are experiments on MNIST in the appendix, however the NMI in those experiments is calculated using non-overlapping clusters.


**Minor points & typos**
- Line 102: "Knadekar et al. Khandekar et al. (2014)" repetition, can be avoided using \citet for the citation. Same for Orecchia et al on line 108. Also for Nepusz et al and Nicosia et al on lines 178 and 179.

- I think it would be slightly better to state the hardness of the objective value right after definition 2.8, so it's immediately clear for the reader (instead of line 272 as it is now)

- A suggestion: Figure 2 does not add much understanding (for me), and I'd prefer to see either Figure 2 or 3 in the main body instead.

**Questions:**

**Q1.)** What is the formal definition of the "width" of a HOC graph?

**Q2.)** What is the hardness of the $k$-HOC problem and $2$-OC problems?

**Q3.)** Does the introduction of the local search heuristics break the approximation guarantees?

**Q4.)** How is the NMI computed for both methods in Figure 1? Section C.1. formally defines NMI for overlapping clusters, however, the reason I ask is that is might follow trivially that the Dasgupta variant in Figure 1 does worse on NMI if it can only assign a single label to each cluster. i.e., the Dasgupta variant returns $k$ non-overlapping clusters.

**Q5.)** Related to above: In Figure 1, does the Dasgupta variant return $k$ non-overlapping or $k$ overlapping clusters?

The rating I've given currently is a 5, however, I am open to increasing my score based on the rebuttal.

---

> ### Author Response · Authors · 2024-11-18
> **Rebuttal**
>
> Thank the reviewer for the valuable comments. We address the concerns one by one.
>
> W1) Sorry for non-intuitive explanations. We say that “although $X$ is nominally a subset of $Y$ , they are possibly unrelated in meaning and clustered by different mechanism”, we mean that even if $X$ is a subset of $Y$, there is not necessarily a directed edge $(X,Y)$ on the HOC graph, because although syntactically we have $X \subseteq Y$, semantically in practice, they may have unrelated meanings from two different systems that are organized by different mechanisms.
>
> On line 191, we can observe that $\alpha_{X,Y}$ in the formula above is defined recursively by layers. The alternate definition afterwards is formulated directly for all pairs $X$ and $Y$. That is, $\alpha_{X,Y}$ is the multiplication of all belonging factors of all parent-child pairs along the unique path from $X$ to $Y$. "keeps the same for the other two cases" means that $\alpha_{X,Y}$ has the same values in the cases of “$X=Y$” and “otherwise” as before, respectively.
>
> In the remark starting on Line 240, the “width” of an HOC graph is defined to be the length of the longest anti-chain of the HOC graph. We say that “it can be considered as a set of key clusters that are close to the leaves on the HOC graph", we seek to capture the “minimal” incomparable clusters. Intuitively, as the clusters aggregate, the cluster level gets high, and the number of incomparable clusters decreases. Consider a non-overlapping HC tree, our definition of width means the number of bottom and smallest clusters, which are the key ones that contain the leaves directly. Similarly, the width of an HOC graph measures the number of the lowest incomparable key clusters that contain the leaves.
>
> W2) Please note that $k$-HOC problem does not restrict to have at most $k$ clusters/nodes. Here $k$ is the restricted width. To be sure, our approximation factor for $k$-HOC is built on that for $2$-OC, but our analysis for $2$-OC is original. In the theoretical study of HC, for both primal and dual problems, a good bipartition is usually the foundation of hierarchical structure. We think HOC is no exception. Regard to the feeling that the dual version is a bit easier to prove for HC, maybe it is due to the hardness of bipartition problem, e.g., sparsest cut, min-bisection, etc., which hinders people to get a constant guarantee for the primal problem of HC. On the contrary, we have constant guarantee for the dual variant.
>
> W3a) Yes, after speed-up for the scalability, the original approximation algorithm (Alg. 2) is not “formally” compared. We think it is acceptable since the speed-up algorithm also follows the local search heuristics (note that Alg. 2 is also a local search). At least, the local search strategy has been verified to be effective and efficient.
>
> W3b) $k$-HOC experiments are only performed on synthetic data because there is no proper real data that has ground truth of HOC. The results corresponding to Dasgupta’s cost in Figure 1 are non-overlapping. The purpose is to demonstrate that compared with using non-overlapping HC cost, our local search algorithm can really benefit from our HOC cost when allowing overlap. Meanwhile, the running time is competitive.
>
> W3c) There is no suitable real data that has manifest ground truth for overlapping clustering, which is the main obstacle in real data evaluation. The experiments on MNIST are devoted to showcase the overlapping data points in our results of image classification. The NMIs in Table 5 are calculated for OC (according to its definition in Section C.1), in which the ground truths are non-overlapping, while our results are overlapping.
>
> Minor points \& typos: We thank the reviewer for the advice in organization, format and figure use.
>
> Q1) See the response to W1).
>
> Q2) The hardness of $k$-HOC and 2-OC problems are still open. Due to the complicated form of cost function and the hybrid structure, we don’t think it is easy, and this direction is of independent interest.
>
> Q3) The guarantee of the speed-up version of our algorithm is actually unknown. It is usually a predicament that an algorithm with strict theoretical guarantee is unable to compete with heuristic algorithms or popular solvers. However, we also think that it is interesting to analyze our heuristics for speed-up.
>
> Q4) See the response to W3b).
>
> Q5) The Dasgupta variant return $k$ non-overlapping clusters. See the response to W3b).

---

> > ### Comment · Reviewer_ZUCp · 2024-11-19
> >
> > I thank the authors for their response and clarifications. I have a couple of follow-up questions/points:
> >
> > 1. Regarding the definition of the width: I thank the authors for their clarification. Could you please change the pdf and add the formal definition? The way it is written up currently it is defined informally half-way through a remark. I think this is important since width is used in some of the theorem/lemma statements.
> >
> > 2. Regarding W3b and Q4, I think at the moment this experiment does not highlight much. As your algorithm returns overlapping clusters, and the Dasgupta variant doesn't, it isn't surprising that your algorithm achieves much better overlapping NMI scores. Are there any experiments that compare against other OC algorithms?

---

> > > ### Author Response · Authors · 2024-11-20
> > > **Response to follow-up questions**
> > >
> > > We thank the reviewer for the timely response.
> > >
> > > 1. Sure, thank you for your suggestion. We will define it formally in the preliminaries and clarify its intuition and significance when we define $k$-HOC problem.
> > >
> > > 2. The effectiveness of our method is evaluated by the high NMIs shown by the subfigures in the 3rd column. We can see that most NMIs are above 0.9, whenever the graph is dense or sparse. We know that it is better to have more baseline OC methods. We can find three OC methods, all of which have been referred to in our related work. The first one is Orecchia et al. (2022) that has been included as the baseline for scalability evaluation. But it is fit for 2-OC only. The second one is Jeantet et al. (2020) that works for vector data and dissimilarity metric, for example, the distance of two nodes. Moreover, it cannot control the number of clusters and the length of anti-chain (according to our definition), which makes it difficult to compare its result with a ground truth that has limited number of clusters. The third one is Gama et al. (2018) that works also dissimilarity metric, and no source code is provided. There is no guarantee for the two latter methods. Although we can define a dissimilarity metric on a graph, for example, by node distance or graph embedding, we think that it is a bit out of scope of our present work.
> > >
> > > References:
> > >
> > > Lorenzo Orecchia, Konstantinos Ameranis, Charalampos Tsourakakis, and Kunal Talwar. Practical almost-linear-time approximation algorithms for hybrid and overlapping graph clustering. In International Conference on Machine Learning, pp. 17071–17093. PMLR, 2022.
> > >
> > > Ian Jeantet, Zoltán Miklós, and David Gross-Amblard. Overlapping hierarchical clustering (OHC). In IDA, volume 12080 of Lecture Notes in Computer Science, pp. 261–273. Springer, 2020.
> > >
> > > Fernando Gama, Santiago Segarra, and Alejandro Ribeiro. Hierarchical overlapping clustering of network data using cut metrics. IEEE Trans. Signal Inf. Process. over Networks, 4(2):392–406, 2018.

---

> > > > ### Comment · Reviewer_ZUCp · 2024-11-20
> > > >
> > > > Thank you for responding again.
> > > >
> > > > 1. You are allowed to edit the pdf for the submission. Can you please include the formal definition of the width? Again, given that it is used in some lemma/theorem statements, it should be included as a formal definition. At the moment it is defined (rather vaguely) in the middle of a remark.
> > > >
> > > > 2. Thank you for clarifying. Given the points I made before, and points raised by other reviewers, I think the experimental section is still slightly weak in this work.

---

> > > > > ### Author Response · Authors · 2024-11-25
> > > > > **Rebuttal**
> > > > >
> > > > > Thank the reviewer for responding. We have updated our new version. Due to the very constrained space, we cannot have a bold font definition for the width, but we have indeed defined it formally in the Preliminaries. We also have clarified its intuition and significance when we define k-HOC problem. We have included the algorithm of Jeantet et al. (2020) as a new baseline. Please refer to our new version and comments for more information.

---

> > > > > > ### Comment · Reviewer_ZUCp · 2024-11-27
> > > > > >
> > > > > > I thank the authors for their response and the work they put into revising the manuscript.
> > > > > >
> > > > > > - In my opinion the write-up has improved significantly (especially the description of the HOC cost function). The explanations of the different concepts such as edge-to-node and node-to-node belonging factors is much clearer.
> > > > > >
> > > > > > - I appreciate the inclusion of the comparison to Jeantet et al (2020), which gives a better picture on how the proposed algorithms compare against other works. It's still difficult to assess how these algorithms would perform on real-world data, however that is maybe due to the lack of a good ground truth dataset. As a remark: Figure 2 is not formatted nicely, and can only be viewed when zooming in a lot. However this can be fixed later (I don't expect the authors to do it before the end of the pdf revision deadline).
> > > > > >
> > > > > >  Taking everything into account, I've increased the score of my review.

---

> > > > > > > ### Author Response · Authors · 2024-11-28
> > > > > > > **Thanks.**
> > > > > > >
> > > > > > > Thank the reviewer for raising the score. Maybe it is better to reduce the scale of the graph in Figure 2 for better visualization. Thanks for the advice.

---

### Official Review · Reviewer_6Krd · 2024-11-02

**Soundness:** 2
**Presentation:** 3
**Contribution:** 2
**Rating:** 3
**Confidence:** 3

**Summary:**

The paper presents the problem of hierarchical overlapping clustering (HOC) for graph data, where clusters may overlap and form hierarchies. The paper presents a cost function for the problem, characterises it formally, and introduces an algorithm that builds on a restricted 2OC version of the problem. It presents lower and upper bounds, as well as approximate versions with guarantees. It presents an  empirical comparison with an existing 2OC solution.

**Strengths:**

S1. The paper is generally well-written and the structure is easy to follow.
S2. The study of hierarchical overlapping clusters on graphs is an interesting potential contribution.
S3. The formalisation and theoretical analysis is adequate.

**Weaknesses:**

W1. The paper should be revised to more clearly present the differences in the problem definition to existing problems OC and HC, respectively.

W2. The experimental results focus only on 2-OC, which is a special case of the proposed general k-HOC, that was already studied in existing work.
An example visualisation of 4-HOC clustering is presented. As the result does not indicate a hierarchy, comparison with existing OC approaches would be in order.
Experiments should demonstrate the benefits of HOC over standard OC for k > 2.

W3. The runtime results in the experiments are not sufficiently described. This is a major concern as the evaluation only compares against a single competitor, by including results from the competitor paper, without re-running. This means that it is unclear whether differences in runtime may be due to implementation details or evaluation framework choices. In particular, the description of the implementation of Orecchia et al. 2022 seems to suggest that it is not optimised for efficiency, meaning that the observed runtime benefits may not necessarily be due to differences in method.

W4. The benefits of the hierarchical aspect of the proposed method are unclear. In the experimental evaluation, the hierarchy is disregarded for calculating NMI, and the visualisation does not present hierarchical results, The paper should provide specific examples or experiments that demonstrate the advantages of the hierarchical structure in HOC compared to flat OC.

W5. The paper should clarify early on (ideally in title, but certainly in abstract or introduction) that the clustering targets graph data.

W6. There are some minor language issues, and the paper would benefit from a strong motivation on the use of the cluster tree results.
Some vague descriptions should be rephrased e.g. "not so many nodes for migration", "we move all movable nodes."

**Questions:**

Please clarify whether Orecchia et al. 2022 runtime results may be the result of their non-optimised implementation but not due to their method as such.

---

> ### Author Response · Authors · 2024-11-18
> **Rebuttal**
>
> Thank the reviewer’s valuable comments. Let us address the concerns one by one.
>
> W1. We provide the formal definitions of cost functions of HC and our HOC in Line 73 (Eq. (1)) and Line 226 (Def 2.8), respectively, and point out that the latter is a generalization from HC by assigning a belonging factor for each minimal common ancestor of each edge. All essential ingredients in our cost function are properly defined in advance. We also provide the cost function for OC in Lines 112-116, which is obviously different from our cost function. Sorry for not having much space to present the differences more.
>
> W2. An evaluation on HOC need an HOC graph model. We have not designed such a model specifically, but utilized the standard OSBM model. The new model design is not the focus of our experiments. We only want to intuitively demonstrate the result of our method by picture.
>
> W3. A major contribution of Orecchia et al. 2022 is the practical nearly linear time complexity of their approximation algorithm. They use the seminal work of Chen et al. (2022) to compute max-flow in theory, but use the HIPR implementation (Cherkassky et al., 1994) of the push-relabel method in experiments. Efficiency is a major contribution in their experiments, and thus we have reasons to believe that they have optimized their implementation. Even if that is not so, please note the huge gap between the runtime of their method and ours demonstrated in Table 1, especially when we consider the difference of experimental operating environments, theirs on a server with 24 Cores and 128GB memory, while ours on a personal computer. This just reflects the value of our cost function (quite different from theirs) and method, simple, easy to implement, and highly efficient.
>
> W4. As we mentioned in the response to W2, the hierarchical aspect should be evaluated by a new HOC graph model. Here, we just use the standard OSBM model for a demonstration. Note that we achieve a high NMI=0.92 on the second level, which implies that the overlapping partition on the first level (as we describe in Line 1452, one cluster of red and yellow, the other one of green and purple) is also accurate.
>
> W5. We agree, and will clarify it early.
>
> W6. Thanks. We will consider our wording carefully.
>
> References:
>
> Li Chen, Rasmus Kyng, Yang P. Liu, Richard Peng, Maximilian Probst Gutenberg, Sushant Sachdeva. Maximum flow and minimum-cost flow in almost-linear time. FOCS, 612-623, 2022.
>
> Cherkassky, B. V., Goldberg, A. V., and Radzik, T. Shortest paths algorithms: theory and experimental evaluation. SODA, 516–525, 1994.

---

> > ### Comment · Reviewer_6Krd · 2024-11-23
> >
> > W1. Indeed, the manuscript gives adequate definitions of HOC, HC, and OC, but lacks a discussion of their differences, in particular in regards to the clusters obtained. How is the generalisation to HOC beneficial?
> > W2. It is difficult to assess the quality of the clustering produced under the proposed HOC model without empirical evaluation of real world clustering results or convincing examples of the benefit of the hierarchies obtained. Even in the absence of a HOC graph model, an evaluation similar to the one in Fig. 2 seems possible also for competitors?
> > W3. I appreciate the effort in conducting empirical evaluation, but the lack of direct experimental comparison in the same experimental setup makes it difficult to assess the relative runtime behaviour.
> > W4. The fact that you get high NMI values is a good indication, but does not provide any information on behaviour relative to state-of-the-art; i.e., is this an easy setting where many (hierarchical) algorithms would obtain high NMI values, or does your approach outperform alternatives?
> > W5, W6. Thank you.

---

> > > ### Author Response · Authors · 2024-11-25
> > > **Rebuttal**
> > >
> > > Thank the reviewer’s response. In our opinion, HOC is a fundamental structure in the organization of datasets, which has also been supported by other work such as Jeantet et al. (2020). So, we didn’t spend to much space on its significance in our paper. We have updated our new version, which has included the algorithm OHC proposed by Jeantet et al. (2020) as a baseline. But its performance is not satisfactory. We also have clarified that our clustering targets graph data in the abstract and at the beginning of the second paragraph. Please refer to our new version and comments for more information.

---

> > > > ### Comment · Reviewer_6Krd · 2024-12-02
> > > >
> > > > Thank you for the revised version, with the clarifications, and a comparison with OHC.
> > > > Thank you for clarifying early on that your work targets graph data. HOC is potentially an interesting concept for graphs, but a clear motivation and differentiation to HC and OC seems relevant, in particular given that e.g. OHC assumes point data.
> > > > I understand that OHC has scalability issues, even though it would be interesting to allow OHC to run for more than just one hour.
> > > > It is unclear to me whether the spectral embedding input to OHC is adequate, as several results have an NMI of zero. Given the relatively small scale of the example data, these results deserve some explanation.

---

> ### Author Response · Authors · 2024-12-03
> **Rebuttal**
>
> Thank you for your response. On the motivation of HOC, we think that there is no essential difference between point data and graph data. Of course, we are willing to give a more concrete example to clarify the significance of HOC on graphs if we have more space.
>
> Regarding the NMI of zero by the OHC’20 algorithm, it results from the “seemly” total misclassification of the four clusters on the first level. Note that even you have a good clustering for the four ground-truth clusters on the second level, you possibly achieve an NMI of zero on the first level. Consider four equal-sized clusters $A$, $B$, $C$, $D$ on Level 2, and the ground truth on Level 1 are two clusters $A\cup B$ and $C\cup D$. Even you achieve a perfect NMI of $1$ on Level 2, if you have a wrong clustering of $A\cup C$ and $B\cup D$ (while keeping the four intact clusters), then the NMI on Level 1 is just zero! It is actually not a serious mistake that you have misclustered the four clusters on the first level maybe only because of a slight difference in the density of linkage among the clusters, but the NMI is very low. This is also the reason why our HOC algorithm has low NMIs (less than $0.7$) on Level 1 of the first two random graphs, but it has high NMIs (more than $0.9$) on Level 2. In the 5 trials on both graphs, we have one trial each that makes a mistake on the first level and gets an NMI of almost zero. This also causes large standard deviations on these two graphs.

---

> > ### Comment · Reviewer_6Krd · 2024-12-03
> >
> > Thank you for your response. The motivation of a hierarchical overlapping clustering on graph data instead of point data remains unclear unfortunately.
> > Thank you also for the NMI example. It clarifies an important issue with the experiments in that the resulting NMI values are not necessarily indicative of the performance of the method, but rather reflect (possibly small changes in) the ground truth clusters.

---

> ### Author Response · Authors · 2024-12-03
> **Rebuttal: on the significance of HOC**
>
> Thank the reviewer. Let us give a practical scenario of HOC. Consider a coauthorship network, in which nodes represent authors and edges represent coauthorship. Then the bottom level clusters are cliques that correspond (although maybe not exactly) to the article that coauthored by a group of people. Since a person is likely to coauthor with many others for many times, such cliques must overlap. However, the granularity of article level is not high enough. Imagine the article level. There must be articles that are written on the same topic (e.g., language model, GNN, clustering, etc.) and include the same author(s). This similarity of articles can be indicated by coauthorship also. Certainly, the topics can overlap (e.g., the article that uses GNN for clustering belongs to the two clusters of GNN and clustering). This forms an upper level of overlapping clustering structure. Moreover, topics belongs to higher level areas (e.g., deep learning, combinatorial optimization, approximation algorithm, etc.), and overlaps happen when a topic belongs to more than one area (e.g., clustering belongs to combinatorial optimization and approximation algorithm, etc.). More widely, (higher) research areas overlap due to interdisciplinary study (e.g., bioinformatics belongs to TCS and biology). This demonstrates a classic HOC structure, and looking at any single perspective of HC or OC alone has its own limitation. We believe that similar story also happens in many other practical scenarios such as social networks, world wide webs, e-commerce websites, etc, which indicates the significance of HOC study on graphs.

---

### Official Review · Reviewer_N9JQ · 2024-11-04

**Soundness:** 2
**Presentation:** 3
**Contribution:** 4
**Rating:** 6
**Confidence:** 3

**Summary:**

The authors introduce a novel cost function for hierarchical overlapping clustering and methods for approximating an optimal solution based on local search.

**Strengths:**

S1) Clusterings that are hierarchical and at the same time overlapping are an important research area. The authors suggest a fitting and coherent solution for a cost function that is consistent with existing research.

S2) The paper is written well, the authors explain their methods and give the intuition behind them.

S3) Supported by theoretic proofs.

**Weaknesses:**

W1) Experimental evaluation should contain more methods than just one baseline competitor.

W2) I'm also missing a more extensive evaluation of the method regarding the datasets. How does your cost function perform for different numbers of clusters, different percentages of overlap, and different models of clusters (centroid-based vs arbitrarily shaped)?

W3) The only baseline results are taken from the original paper they were reported in as the authors were "not able to compile correctly their codes published online". Usually, one would reimplement the method based on the publication s.t. it is possible to compare the methods for more experiments.

Small note: the citation format is sometimes a bit off, especially when the publication is referred to by the authors' names (e.g., lines 094 - 102 : try that the authors's names are not just repeated twice)

**Questions:**

Q1) How is the NMI computed in Figure 1? As far as I know, there is no standard way to include the concepts of overlapping and hierarchical clustering in the evaluation with NMI, so how did you measure it?

---

> ### Author Response · Authors · 2024-11-18
> **Rebuttal**
>
> Thank the reviewer’s valuable comments. Let us address the concerns one by one.
>
> W1) There are indeed several methods for overlapping clustering, as what we have introduced in the related work. However, except Orecchia et al. (2022) that we choose as the only baseline competitor, no work provides cost function and guarantee on performance, while based only on some heuristics like density or cut metrics during the computing processes. This is why we, and also Orecchia et al. (2022), haven’t contained them as baselines.
>
> W2) Yes, a more extensive evaluation of method is preferable for a new concept. In our paper, cluster number, overlapping percentage, density of clusters, balance of clusters, etc., are all factors that are worth evaluation. We once have a comprehensive evaluation on our 2-OC algorithm that has an excellent performance, but have not presented it in our submission. However, in present version, we mainly focus on the accuracy and scalability of our method. Thanks for your advice to have more experiments.
>
> W3) We are usually in a predicament when we try to involve a baseline method but the source code is hard to compile correctly, especially when it has a complicated computing process for theoretical guarantee. That is the case for Orecchia et al. (2002)’s work. Neither debug nor reimplementation are easy. However, we can see the huge gap between the scalabilities of their work and ours demonstrated in Table 1, especially when we consider the difference of experimental operating environments, theirs on a server with 24 Cores and 128GB memory, while ours on a personal computer. This just reflects the value of our method, simple, easy to implement, and highly efficient.
>
> Small note: Thanks, we will use a proper citation format in future version.
>
> Q1) The NMI for overlapping clustering is proposed by Lancichinetti et al. (2009), and we also give the formal definition in Appendix C.1 (as we state in Lines 445-446).
>
> References:
>
> Lorenzo Orecchia, Konstantinos Ameranis, Charalampos Tsourakakis, and Kunal Talwar. Practical nearly-linear-time approximation algorithms for hybrid and overlapping graph clustering. In International Conference on Machine Learning, pp. 17071–17093. PMLR, 2022.
>
> Andrea Lancichinetti, Santo Fortunato, and János Kertész. Detecting the overlapping and hierarchical community structure in complex networks. New Journal of Physics, 11(3):033015, 2009.

---

> > ### Comment · Reviewer_N9JQ · 2024-11-21
> >
> > Thanks for your answers.
> >
> > W1) Would it be possible to include those methods, nevertheless? It would be interesting to see how well the other algorithms perform, e.g., using your cost function or external labeling.
> >
> > W2) If you have those experiments already, could you make them available in the revised version? Or at least in the supplementary material? There is enough space for that. Otherwise, could you summarize their results?
> >
> >
> > Q1) Thanks for clarifying. I think there is a typo in your formula in line 1359 (should be $x_i \in X$, not $x_i \in x$)

---

> > > ### Author Response · Authors · 2024-11-25
> > > **Rebuttal**
> > >
> > > Thank the reviewer’s response. We have updated our new version, and the algorithm proposed by Jeantet et al. (2020) has been included as a new baseline. Its performance is not so good. Please refer to our new version and comments for more information.
> > >
> > > Thanks for pointing out the typo.

---

### Author Response · Authors · 2024-11-25
**New version updated.**

Hope this update is not too late.

We have updated our new version. It contains several major modifications that address the concerns of the reviewers. We list them as follows.

(1) We have included the algorithm, named as OHC’20, proposed by Jeantet et al. (2022) as a new baseline. OHC’20 is a heuristic algorithm for hierarchical overlapping clustering for vector data. To fit to graph clustering, we take the spectral embedding (just like what people do in spectral clustering) first, and then feed it to OHC’20. However, OHC’20 needs to deal with the adjacency matrix that includes all-pair distance of data points, which restricts seriously the scalability of this algorithm. So, for a graph of size at least $10^4$, we didn’t wait until its termination within one hour, and its experiment on large graphs aborts. Moreover, it cannot control the hierarchy number and cluster number. In NMI evaluation, we pick the level that has the best NMI, but the results are not satisfactory also.

(2) We have adjusted the structure of our manuscript. We move the example that consists of two overlapping triangles to the main body as a running example, and all important definitions on HOC graphs have corresponding explanations on it. Due to the space limit, we have moved all contents on 2-OC-P and visualization to the appendices.

(3) We have changed the settings of OSBM in our 4-HOC experiments such that there are two hierarchies in the ground-truth graphs. We set two values of probability for inter-links between underlying clusters. Accordingly, the NMI evaluation and visualization are conducted on these two levels separately.

(4) We have fixed unclear points, citation formats and typos that raised by the reviewers.

Thank all the reviewers for the hard working and valuable advice.

---

### Meta-Review · Area_Chair_ZK5V · 2024-12-17

**Metareview:**

This paper tackles the problem of hierarchical overlapping clustering (HOC) for graph data, where clusters can overlap and form hierarchical structures. The paper introduces a new cost function for HOC, inspired by the Dasgupta cost function for hierarchical clustering, and provides a constant-factor approximation algorithm for the dual objective.

The reviewers find the problem of HOC interesting and relevant, and appreciate the introduction of a new cost function with desirable properties. The theoretical results, particularly the approximation algorithm for the dual objective, are acknowledged as a contribution to the field.

However, reviewers also raise some significant concerns:

- Dual Objective Approximation: The main theoretical result focuses on approximating the dual objective, which does not provide a direct guarantee on the primal objective value. This limits the practical implications of the theoretical guarantee.
- Unconvincing Experimental Evaluation: The experimental evaluation is not comprehensive enough and does not provide strong evidence for the effectiveness of the proposed method.

Recommendation:

While the paper introduces a novel problem and provides some theoretical analysis, the reviewers agree that it does not meet the bar for acceptance at ICLR.

**Additional Comments On Reviewer Discussion:**

The paper has been discussed in the rebuttal phase and the reviewers agree on the final decision

---

### Decision · Program_Chairs · 2025-01-22

Reject